# THE DISCRIMINATIVE JACKKNIFE: QUANTIFYING PREDICTIVE UNCERTAINTY VIA HIGHER-ORDER INFLUENCE FUNCTIONS

## ABSTRACT

Deep learning models achieve high predictive accuracy in a broad spectrum of tasks, but rigorously quantifying their predictive uncertainty remains challenging. Usable estimates of predictive uncertainty should (1) *cover* the true prediction target with a high probability, and (2) *discriminate* between high- and low-confidence prediction instances. State-of-the-art methods for uncertainty quantification are based predominantly on Bayesian neural networks. However, Bayesian methods may fall short of (1) and (2) — i.e., Bayesian credible intervals do not guarantee frequentist coverage, and approximate posterior inference may undermine discriminative accuracy. To this end, this paper tackles the following question: *can we devise an alternative frequentist approach for uncertainty quantification that satisfies (1) and (2)?*

To address this question, we develop the *discriminative jackknife* (DJ), a formal inference procedure that constructs predictive confidence intervals for a wide range of regression models, is easy to implement, and provides rigorous theoretical guarantees on (1) and (2). The DJ procedure uses higher-order *influence functions* (HOIFs) of the trained model parameters to construct a jackknife (leave-one-out) estimator of predictive confidence intervals. DJ computes HOIFs using a recursive formula that requires only oracle access to loss gradients and Hessian-vector products, hence it can be applied in a *post-hoc* fashion without compromising model accuracy or interfering with model training. Experiments demonstrate that DJ performs competitively compared to existing Bayesian and non-Bayesian baselines.

## 1 INTRODUCTION

Deep learning models have achieved state-of-the-art performance on a variety of supervised learning tasks, and are becoming increasingly popular in various application domains (LeCun et al. (2015)). A key question often asked of such models is "*Can we trust this particular model prediction?*" This question is highly relevant in high-stakes applications where the model is used to guide and inform critical decision-making — examples of such applications include: medical decision support, autonomous vehicle control, and financial forecasting (Amodei et al. (2016)). Despite their impressive predictive accuracy, rigorously quantifying the predictive uncertainty of deep learning models is a challenging and yet an unresolved problem (Gal (2016); Ovadia et al. (2019)).

Actionable estimates of predictive uncertainty are ones that (1) *cover* the true prediction target with high probability, and (2) *discriminate* between high- and low-confidence predictions. (Figure 1 depicts a pictorial visualization for the coverage and discrimination requirements.) The first requirement, frequentist coverage, is especially relevant in applications where predictive uncertainty is incorporated in a decision-theoretic framework (e.g., administering medical treatments (Dusenberry et al. (2019)), or estimating value functions in model-free reinforcement learning (White & White (2010))). The second requirement, discrimination, is crucial for auditing model reliability (Schulam & Saria (2019)), detecting dataset shifts and out-of-distribution samples (Barber et al. (2019a)), and actively collecting new training examples for which the model is not confident (Cohn et al. (1996)).

Existing methods for uncertainty quantification are based predominantly on Bayesian neural networks (BNNs), whereby predictive uncertainty is evaluated via posterior credible intervals (Welling & Teh (2011); Hernández-Lobato & Adams (2015); Ritter et al. (2018); Maddox et al. (2019)). However,

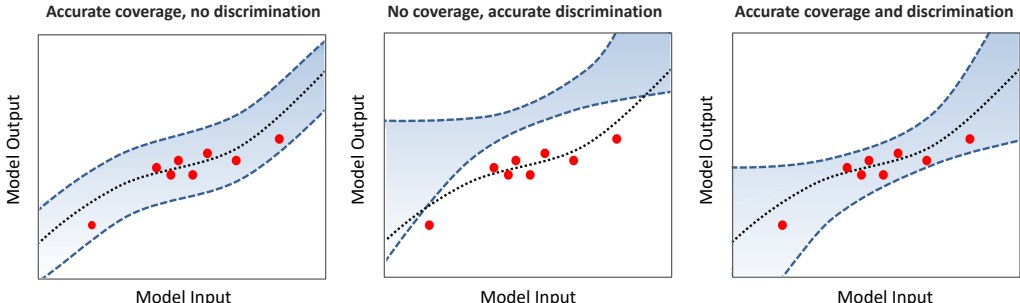

Figure 1: **Pictorial depiction for coverage and discrimination in uncertainty estimates.** Red dots correspond to training data and dotted black line corresponds to the target function. Confidence intervals are visualized as shaded blue regions, and dotted blue lines are the upper and lower confidence bounds. The left panel shows a confidence interval that perfectly covers the data points, but does not discriminate high-confidence predictions (regions with dense training data) and low-confidence ones (regions with scarce training data). The middle panel shows a confidence interval with a width proportional to the density of training data, but does not cover any data point. The right panel shows a confidence interval that satisfies both coverage and discrimination requirements.

BNNs require significant modifications to the training procedure, and exact Bayesian inference is computationally prohibitive in practice. Approximate dropout-based inference schemes (e.g., Monte Carlo dropout (Gal & Ghahramani (2016)) and variational dropout (Kingma et al. (2015))) have been recently proposed as computationally efficient alternatives. However, Bayesian inference in dropout-based models has been shown to be ill-posed, since the induced posterior distributions in such models do not concentrate asymptotically (Osband (2016); Hron et al. (2017)), which jeopardizes both the coverage and discrimination performance of the resulting credible intervals. Moreover, even with exact inference, Bayesian credible intervals generally do not guarantee frequentist coverage (Bayarri & Berger (2004)). Alternative non-Bayesian methods have been recently developed — mostly based on ad-hoc ensemble approaches (Lakshminarayanan et al. (2017)) — but formal frequentist methods with rigorous theoretical guarantees are still lacking.

**Summary of Contributions.** In this paper, we develop a formal procedure for constructing frequentist (pointwise) *confidence intervals* on the predictions of a broad class of deep learning models. Our method — which we call the *discriminative jackknife* (DJ) — is inspired by the classic jackknife leave-one-out (LOO) re-sampling procedure for estimating variability in statistical models (Miller (1974); Efron (1992)). In order to ensure both frequentist coverage and discrimination, DJ constructs feature-dependent confidence intervals using the LOO local prediction variance at the input feature, and adjusts the interval width (for a given coverage probability) using the model's average LOO error residuals. Whereas the classic jackknife satisfies neither the coverage nor the discrimination requirements (Barber et al. (2019b)), DJ provides strong theoretical guarantees on both (i.e., DJ generates confidence intervals resembling those in the rightmost panel of Figure 1 with high probability).

Central to the DJ procedure is the usage of *influence functions* — a key concept in robust statistics and variational calculus (Efron (1992); Cook & Weisberg (1982); Hampel et al. (2011)) — in order to estimate the parameters of models trained on LOO versions of the training data, without the need to exhaustively re-train the model for each held-out data point. That is, using the *von Mises* expansion (Fernholz (2012)) — a variant of Taylor series expansion for statistical functionals — we represent the (counter-factual) model parameters that would have been learned on LOO versions of the training dataset in terms of an infinite series of higher-order influence functions (HOIFs) for the model parameters trained on the complete dataset. We show that a truncated von Mises expansion (with a finite number of HOIFs) is sufficient to preserve the guarantees on coverage and discrimination. To enable efficient computation of the von Mises expansion, we derive a recursive formula that links HOIFs with lower-order influence functions and the partial derivatives of the model loss function.

Comprehensive experimental evaluation demonstrates that the DJ procedure performs competitively compared to both Bayesian and non-Bayesian methods. In particular, we found that the DJ procedure outperforms Bayesian neural networks with respect to discriminative performance, and provides better frequentist coverage compared to existing non-Bayesian methods, while being significantly less computationally expensive than both classes of methods.

## 2   PRELIMINARIES

**Learning Setup.** We consider a standard supervised learning setup with $(\boldsymbol{x}, y)$ being a feature-label pair, where the feature $\boldsymbol{x}$ belongs to a $d$-dimensional *feature space* $\mathcal{X} \subseteq \mathbb{R}^d$, and $y \in \mathbb{R}$. A model is trained to predict $y$ using a dataset $\mathcal{D} \triangleq \{(\boldsymbol{x}_i, y_i)\}_{i=1}^n$ of $n$ training examples which are drawn i.i.d from an unknown distribution $\mathbb{P}$. Let the learning model be denoted as $f(\boldsymbol{x}; \theta) : \mathcal{X} \to \mathcal{Y}$, where $\theta \in \Theta$ are the model parameters, and $\Theta$ is the parameter space. The learned model parameters $\hat{\theta} \in \Theta$ are obtained by solving the optimization problem $\hat{\theta} = \arg\min_{\theta \in \Theta} L(\mathcal{D}, \theta)$, with a loss function

$$L(\mathcal{D}, \theta) \triangleq \tfrac{1}{n} \sum_{i=1}^n \ell(y_i, f(\boldsymbol{x}_i; \theta)), \tag{1}$$

where we fold in any regularization terms into $\ell(.)$. We do not pose any assumptions on how the loss function in (1) is optimized. The model $f(\boldsymbol{x}; \theta)$ can be any neural network variant (e.g., feed-forward, convolutional, etc). Throughout the paper, we limit our discussion to the regression task; extending our framework to classification tasks is an interesting direction for future work.

**Uncertainty Quantification.** The predictions issued by the trained model (with learned parameters $\hat{\theta}$) are given by $f(\boldsymbol{x}; \hat{\theta})$. Our main goal is to obtain estimates for the uncertainty in the model predictions, expressed through a pointwise *confidence interval* $\mathcal{C}(\boldsymbol{x}; \hat{\theta})$ defined as follows:

$$\mathcal{C}(\boldsymbol{x}; \hat{\theta}) \triangleq [\, f_L(\boldsymbol{x}; \hat{\theta}),\ f_U(\boldsymbol{x}; \hat{\theta}) \,],\ f_L(\boldsymbol{x}; \hat{\theta}) < f(\boldsymbol{x}; \hat{\theta}) < f_U(\boldsymbol{x}; \hat{\theta}),\ \forall \boldsymbol{x} \in \mathcal{X}. \tag{2}$$

The extent of uncertainty in the model's prediction (for a data point with feature $\boldsymbol{x}$) is quantified by the *interval width* $W(.)$ of $\mathcal{C}(\boldsymbol{x}; \hat{\theta})$, defined as $W(\mathcal{C}(\boldsymbol{x}; \hat{\theta})) \triangleq f_U(\boldsymbol{x}; \hat{\theta}) - f_L(\boldsymbol{x}; \hat{\theta})$. Wider intervals imply less confidence, and vice versa. For $\mathcal{C}(\boldsymbol{x}; \hat{\theta})$ to be usable, it has to satisfy the following requirements:

---

(i) *Frequentist coverage:* The confidence interval $\mathcal{C}(\boldsymbol{x}; \hat{\theta})$ covers the true prediction target with an arbitrarily prespecified coverage probability of $(1 - \alpha)$, $\alpha \in (0, 1)$, i.e.,

$$\mathbb{P}\left\{ y \in \mathcal{C}(\boldsymbol{x}; \hat{\theta}) \right\} \geq 1 - \alpha,$$

where the probability is taken with respect to a (new) test point $(\boldsymbol{x}, y)$ as well as with respect to the training data $\mathcal{D}$ (Lawless & Fredette (2005); Barber et al. (2019b)).

(ii) *Discrimination:* The confidence interval $\mathcal{C}(\boldsymbol{x}; \hat{\theta})$ is wider (on average) for test points with less accurate predictions (Leonard et al. (1992)), i.e., for the test points $\boldsymbol{x}, \boldsymbol{x}' \in \mathcal{X}$, we have

$$\mathbb{E}[W(\mathcal{C}(\boldsymbol{x}; \hat{\theta}))] \geq \mathbb{E}[W(\mathcal{C}(\boldsymbol{x}'; \hat{\theta}))] \iff \mathbb{E}[\ell(y, f(\boldsymbol{x}; \hat{\theta}))] \geq \mathbb{E}[\ell(y', f(\boldsymbol{x}'; \hat{\theta}))],$$

where the expectation $\mathbb{E}[.]$ is taken with respect to the training data $\mathcal{D}$.

---

In the next Section, we develop a formal *post-hoc* procedure that takes the dataset $\mathcal{D}$ and the trained model $f(\boldsymbol{x}; \hat{\theta})$ as inputs, and evaluates an uncertainty estimate $\widehat{\mathcal{C}}(\boldsymbol{x}; \hat{\theta})$ that satisfies both requirements.

## 3   THE DISCRIMINATIVE JACKKNIFE

Before presenting our proposed *discriminative jackknife* (DJ) procedure, we start with a brief recap of the classical jackknife. The jackknife quantifies predictive uncertainty in terms of the (average) prediction error, which is estimated with a leave-one-out (LOO) construction found by systematically leaving out each data point from the training dataset $\mathcal{D}$, and evaluating the error of the re-trained model on the held-out sample. That is, for a target coverage of $(1 - \alpha)$, a naïve jackknife estimates the uncertainty in the prediction for a test point $\boldsymbol{x}$ as (Miller (1974); Efron (1992)):

$$\widehat{\mathcal{C}}_{\alpha,n}(\boldsymbol{x}; \hat{\theta}) = f(\boldsymbol{x}; \hat{\theta}) \pm \widehat{Q}_{\alpha,n}(\mathcal{R}),\ \mathcal{R} = \{r_1, \dots, r_n\}, \tag{3}$$

where $r_i = |\, y_i - f(\boldsymbol{x}_i; \hat{\theta}_{-i}) \,|$ is the error residual on the $i^{th}$ data point, $\hat{\theta}_{-i}$ are the parameters of the model re-trained on the dataset $\mathcal{D} \setminus \{(\boldsymbol{x}_i, y_i)\}$ with the $i^{th}$ point removed, and $\widehat{Q}_{\alpha,n}$ is the $(1 - \alpha)$ quantile for the empirical distribution of elements of the set $\mathcal{R} = \{r_1, \dots, r_n\}$ defined as[1]

$$\widehat{Q}_{\alpha,n}(\mathcal{R}) \triangleq \text{ the } \lceil (1 - \alpha)\,(n + 1) \rceil\text{-th smallest value of the elements of } \mathcal{R}.$$

---

[1]Note that, from the definition of $\widehat{Q}_{\alpha,n}$, it follows that $\widehat{Q}_{\alpha,n}(\mathcal{R}) = \widehat{Q}_{1-\alpha,n}(-\mathcal{R})$.

Albeit intuitive, the naïve jackknife estimate in (3) is not guaranteed to achieve the target coverage of $(1 - \alpha)$ (Barber et al. (2019b)). More crucially, the interval width $W(\widehat{\mathcal{C}}_{\alpha,n}(\boldsymbol{x}; \hat{\theta}))$ is a constant that does not depend on the feature $\boldsymbol{x}$, which renders discrimination impossible, i.e., when applied, the naïve jackknife would result in confidence intervals resembling the leftmost panel in Figure 1.

### 3.1 EXACT CONSTRUCTION OF THE DJ CONFIDENCE INTERVALS

We propose an ameliorated jackknife procedure, the DJ, which addresses the shortcomings of the classical jackknife. We start by defining some notation. Let the sets $\mathcal{V}_n^+(\boldsymbol{x})$ and $\mathcal{V}_n^-(\boldsymbol{x})$ be defined as:

$$\mathcal{V}_n^+(\boldsymbol{x}) = \{\, v_i(\boldsymbol{x}) \mid \forall i,\, v_i(\boldsymbol{x}) \geq 0 \,\}, \text{ and } \mathcal{V}_n^-(\boldsymbol{x}) = \{\, -v_i(\boldsymbol{x}) \mid \forall i,\, v_i(\boldsymbol{x}) < 0 \,\}, \quad (4)$$

where $v_i(\boldsymbol{x}) = f(\boldsymbol{x}; \hat{\theta}) - f(\boldsymbol{x}; \hat{\theta}_{-i})$, $n^+ = |\mathcal{V}_n^+|$, and $n^- = |\mathcal{V}_n^-|$. Our DJ procedure estimates the predictive confidence interval for test point $\boldsymbol{x}$ as follows: $\widehat{\mathcal{C}}_{\alpha,n}(\boldsymbol{x}; \hat{\theta}) = [\, f_L(\boldsymbol{x}; \hat{\theta}), f_U(\boldsymbol{x}; \hat{\theta}) \,]$, where

$$
\begin{aligned}
f_L(\boldsymbol{x}; \hat{\theta}) &= f(\boldsymbol{x}; \hat{\theta}) - \widehat{Q}_{\alpha,n}(\mathcal{R}_n) - \widehat{Q}_{1-\alpha,n^-}(\mathcal{V}_n^-(\boldsymbol{x})), \\
f_U(\boldsymbol{x}; \hat{\theta}) &= f(\boldsymbol{x}; \hat{\theta}) + \underbrace{\widehat{Q}_{\alpha,n}(\mathcal{R}_n)}_{\text{Marginal Error}} + \underbrace{\widehat{Q}_{\alpha,n^+}(\mathcal{V}_n^+(\boldsymbol{x}))}_{\text{Local Variability}}.
\end{aligned}
\quad (5)
$$

Unlike the traditional jackknife, the construction of the DJ confidence intervals comprises two terms: *marginal prediction error* and *local prediction variability*. The prediction error term, $\widehat{Q}_{\alpha,n}(\mathcal{R}_n)$, uses the LOO residuals to estimate the model's (marginal) generalization error. The prediction variability terms, $\widehat{Q}_{\alpha,n^+}(\mathcal{V}_n^+(\boldsymbol{x}))$ and $\widehat{Q}_{1-\alpha,n^-}(\mathcal{V}_n^-(\boldsymbol{x}))$, quantify the extent to which each training data point impacts the value of the prediction at test point $\boldsymbol{x}$. The prediction error is constant, i.e., does not depend on $\boldsymbol{x}$, hence it only contributes to coverage but does not contribute to discrimination. On the contrary, the local variability terms depend on the test point feature $\boldsymbol{x}$, hence they fully determine the discrimination performance. As we show later in Section 3.4, constructing confidence intervals this way provides strong theoretical guarantees on the achieved frequentist coverage and discrimination.

Figure 2 illustrates the construction of the DJ confidence intervals in (5). The confidence intervals are chosen so that the boundaries of the average error and local variability are exceeded by $\lceil (n+1)(1-\alpha) \rceil$ out of the $n$ LOO samples — these are marked with a star. For the average prediction error term, the width of the resulting boundary is the same for any test data point $\boldsymbol{x} \in \mathcal{X}$. For the local prediction variability term, the width of the boundary depends on the specific value of the feature $\boldsymbol{x}$, and should get wider for less confident prediction instances.

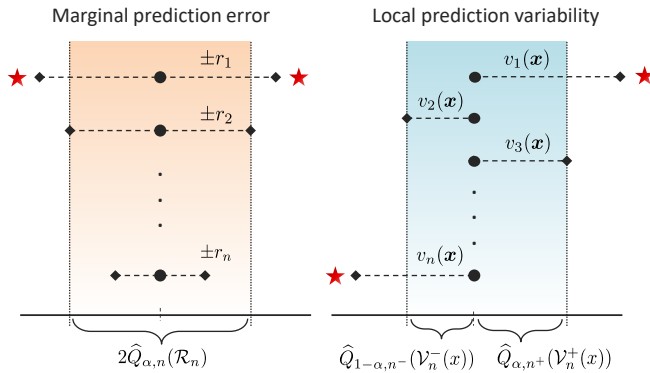

Figure 2: Illustration of the discriminative jackknife.

Based on the confidence intervals in (5), it follows that the DJ interval width is given by

$$W(\widehat{\mathcal{C}}_{\alpha,n}(\boldsymbol{x}; \hat{\theta})) = 2\,\widehat{Q}_{\alpha,n}(\mathcal{R}_n) + \widehat{Q}_{1-\alpha,n^-}(\mathcal{V}_n^-(\boldsymbol{x})) + \widehat{Q}_{\alpha,n^+}(\mathcal{V}_n^+(\boldsymbol{x})). \quad (6)$$

The marginal error and local variability terms in (6) jointly capture two types of uncertainty: *epistemic* and *aleatoric* uncertainties (Gal (2016)). Epistemic uncertainty measures how well the model is matched to the data, and is reducible as the size of training data increases. On the contrary, aleatoric uncertainty is the irreducible uncertainty arising from the inherent sources of complexity in the data, such as label noise or hidden features (Malinin & Gales (2018)). Consistency of the DJ confidence estimates requires that $W(\widehat{\mathcal{C}}_{\alpha,n}(\boldsymbol{x}; \hat{\theta})) \to 0$, i.e., the interval width vanishes, as the size of the training data increases ($n \to \infty$). It follows from (6) that if there is no aleatoric uncertainty, and if the underlying model is stable (i.e., $\lim_{n\to\infty} v_i = 0$) and consistent (i.e., $\lim_{n\to\infty} r_i = 0$), then the interval width $W(\widehat{\mathcal{C}}_{\alpha,n}(\boldsymbol{x}; \hat{\theta}))$ vanishes as $n$ grows asymptotically (more training data is collected).

## 3.2 Efficient Implementation via Higher-Order Influence Functions

Exact computation of the DJ confidence intervals in (5) requires re-training the model $n$ times in order to collect the "perturbed" LOO parameters $\{\hat{\theta}_{-i}\}_{i=1}^{n}$. To scale up the DJ procedure, we propose to use *influence functions* — a classic tool from robust statistics (Huber & Ronchetti (1981); Hampel et al. (2011)) — in order to recover the parameters $\{\hat{\theta}_{-i}\}_{i=1}^{n}$ on the basis of the trained model $f(\boldsymbol{x}; \hat{\theta})$, without the need for repeated re-training. Through our influence function-based implementation, the DJ can be applied in a *post-hoc* fashion, requiring only knowledge of the model loss gradients.

Influence functions enable efficient computation of the effect of a training data point $(\boldsymbol{x}_i, y_i)$ on $\hat{\theta}$. This is achieved by evaluating the change in $\hat{\theta}$ if $(\boldsymbol{x}_i, y_i)$ was up-weighted[2] by some small $\epsilon$, resulting in a new parameter $\hat{\theta}_{i,\epsilon} \triangleq \arg\min_{\theta \in \Theta} L(\mathcal{D}, \theta) + \epsilon \cdot \ell(y_i, f(\boldsymbol{x}_i; \theta))$. The rate of change in $\hat{\theta}$ due to an infinitesimal perturbation $\epsilon$ in data point $i$ is give by the (first-order) influence function, defined as

$$\mathcal{I}_{\theta}^{(1)}(\boldsymbol{x}_i, y_i) = \frac{\partial \hat{\theta}_{i,\epsilon}}{\partial \epsilon} \Big|_{\epsilon=0}. \tag{7}$$

Note that the model parameter $\hat{\theta}$ is a *statistical functional* of the data distribution $\mathbb{P}$. Perturbing the $i^{th}$ training point is equivalent to perturbing $\mathbb{P}$ to create a new distribution $\mathbb{P}_{i,\epsilon} = (1 - \epsilon)\,\mathbb{P} + \epsilon\Delta(\boldsymbol{x}_i, y_i)$, where $\Delta(\boldsymbol{x}_i, y_i)$ denotes the Dirac distribution in the point $(\boldsymbol{x}_i, y_i)$. In this sense, the influence function in (7) operationalizes the concept of derivatives to statistical functionals, i.e., it quantifies the derivative of the learned model parameters $\hat{\theta}$ with respect to the data distribution $\mathbb{P}$.

By recognizing that influence functions are the "derivatives" of the model parameters $\hat{\theta}$ with respect to the data distribution $\mathbb{P}$, we can use a Taylor-type expansion to represent the counter-factual model parameter $\hat{\theta}_{i,\epsilon}$ (that would have been learned from a dataset with the $i^{th}$ data point up-weighted) in terms of the true model parameter $\hat{\theta}$ (learned from the true training dataset $\mathcal{D}$) as follows:

$$\hat{\theta}_{i,\epsilon} = \hat{\theta} + \epsilon \cdot \mathcal{I}_{\theta}^{(1)}(\boldsymbol{x}_i, y_i) + \frac{\epsilon^2}{2!} \cdot \mathcal{I}_{\theta}^{(2)}(\boldsymbol{x}_i, y_i) + \ldots + \frac{\epsilon^m}{m!} \cdot \mathcal{I}_{\theta}^{(m)}(\boldsymbol{x}_i, y_i) + \ldots, \tag{8}$$

where $\mathcal{I}_{\theta}^{(k)}(\boldsymbol{x}_i, y_i)$ is the $k^{th}$ order influence function, defined as $\mathcal{I}_{\theta}^{(k)}(\boldsymbol{x}_i, y_i) = \partial^k \hat{\theta}_{i,\epsilon} / \partial \epsilon^k |_{\epsilon=0}$. The expansion in (8), known as the *von Mises* expansion (Fernholz (2012)), is a distributional analog of the Taylor expansion for a statistical functional. If all of the higher-order influence functions (HOIFs) in (8) exist, then we can recover $\hat{\theta}_{i,\epsilon}$ without re-training the model on the perturbed training dataset. In Section 3.3, we will derive a recursive formula for efficiently computing the HOIFs in (8). Since exact reconstruction of $\hat{\theta}_{i,\epsilon}$ requires an infinite number of HOIFs, we can only approximate $\hat{\theta}_{i,\epsilon}$ by including a sufficient number of HOIF terms from the von Mises expansion. We discuss the impact of this approximation on the coverage and discrimination performance of the DJ in Section 3.4.

Note that the LOO model parameters $\{\hat{\theta}_{-i}\}_{i=1}^{n}$ (required for the construction of the DJ confidence intervals) can be obtained by setting $\epsilon = -1/n$, i.e., $\hat{\theta}_{-i} = \hat{\theta}_{i, \frac{-1}{n}}$, since removing a training point is equivalent to up-weighting it by $-1/n$ in the loss function $L(\mathcal{D}; \theta)$. Thus, by setting $\epsilon = -1/n$ and selecting a prespecified number of HOIF terms $m$ for obtaining the approximate LOO parameters $\hat{\theta}_{-i}^{(m)}$, the DJ confidence intervals can be computed using the steps in Algorithm 1.

---

**Algorithm 1:** The Discriminative Jackknife

**Input** : Learned parameter $\hat{\theta}$, influence order $m$, coverage $\alpha$, training data $\mathcal{D}$, test point $\boldsymbol{x}$

**Output** : Approximate DJ confidence interval $\widehat{\mathcal{C}}_{\alpha,n}^{(m)}(\boldsymbol{x}; \hat{\theta})$

1: **for** $i = 1$ *to* $n$ **do**
2: $\quad$ $\hat{\theta}_{-i}^{(m)} \leftarrow \hat{\theta} - \sum_{k=1}^{m} (n^{-k}/k!) \cdot \mathcal{I}_{\theta}^{(k)}(\boldsymbol{x}_i, y_i)$
3: $\quad$ $r_i \leftarrow \big| Y_i - f(\boldsymbol{x}_i; \hat{\theta}_{-i}^{(m)}) \big|$
4: $\quad$ $v_i(\boldsymbol{x}) \leftarrow f(\boldsymbol{x}; \hat{\theta}) - f(\boldsymbol{x}; \hat{\theta}_{-i}^{(m)})$
5: **end**
6: $f_L(\boldsymbol{x}; \hat{\theta}) \leftarrow f(\boldsymbol{x}; \hat{\theta}) - \widehat{Q}_{\alpha,n}(\mathcal{R}_n) - \widehat{Q}_{1-\alpha,n^-}(\mathcal{V}_n^-(\boldsymbol{x}))$
7: $f_U(\boldsymbol{x}; \hat{\theta}) \leftarrow f(\boldsymbol{x}; \hat{\theta}) + \widehat{Q}_{\alpha,n}(\mathcal{R}_n) + \widehat{Q}_{\alpha,n^+}(\mathcal{V}_n^+(\boldsymbol{x}))$
8: **return:** $\widehat{\mathcal{C}}_{\alpha,n}^{(m)}(\boldsymbol{x}; \hat{\theta}) \leftarrow [\, f_L(\boldsymbol{x}; \hat{\theta}), f_U(\boldsymbol{x}; \hat{\theta}) \,]$

---

[2]Estimators that perturb parameters instead of a LOO construct are known as the *infinitesimal jackknife* (IJ) (Giordano et al. (2018)). A detailed technical background on influence functions is provided in Appendix A.

### 3.3 Computing Higher-Order Influence Functions

How can we compute the influence functions of the von Mises expansion in (8)? The recent work on model interpretability in (Koh & Liang (2017)) has studied the use of influence functions to quantify the impact of each training data point on a model's predictions. In this work, first-order influence functions were computed — using the classical result in (Cook & Weisberg (1982)) — as follows:

$$\mathcal{I}_\theta^{(1)}(\boldsymbol{x}, y) = -H_\theta^{-1} \cdot \nabla_\theta \, \ell(y, f(\boldsymbol{x}, \theta)), \tag{9}$$

where $H_\theta \triangleq \nabla_\theta^2 \, \ell(y, f(\boldsymbol{x}, \theta))$ is the Hessian of the loss function, which is assumed to be positive definite. However, the expression in (9) is only limited to first-order influence functions. In the following Theorem[3], we derive a recursive formula for computing HOIFs of the model parameters $\theta$.

**Theorem 1.** *Let $\ell(.)$ be a differentiable function with a positive definite Hessian matrix $H_\theta$. If $\ell(.)$ is locally convex in the neighborhood of $\theta$, then the $(k+1)$-th order influence function of $\theta$ exists for all $(\boldsymbol{x}, y) \in \mathcal{X} \times \mathcal{Y}$, and for all $k \geq 1$ we have*

$$\mathcal{I}_\theta^{(k+1)}(\boldsymbol{x}, y) = -H_\theta^{-1} \Big( \sum_{m=1}^{k} a_m \cdot g_m \Big( \big\{ \mathcal{I}_\theta^{(j)}(\boldsymbol{x}, y), \nabla_\theta^j \ell(y, f(\boldsymbol{x}, \theta), \nabla_\theta^j L(\mathcal{D}, \theta)) \big\}_{j=1}^m \Big) \Big),$$

*where $a_m$ is a constant, and $g_m$ is a polynomial function, $\forall m \in \{1, \ldots, m\}$.* □

Similar expressions have been arrived at in (Giordano et al. (2019); Debruyne et al. (2008)). We give the precise definition of the constant term $a_m(.)$ and the function $g_m(.)$ — in terms of Bell polynomials — in Appendix B. A key consequence of Theorem 1 is that to compute all HOIFs, we need to evaluate the inverse of the Hessian matrix $H_\theta^{-1}$ only once, hence evaluating HOIFs scales linearly with the number of training examples. With automatic differentiation tools at our disposal (Paszke et al. (2017)), computing the loss gradients (arguments of $g_m(.)$) can be done efficiently.

As it is the case in first-order influence functions, inverting $H_\theta$ is the bottleneck operation in evaluating HOIFs, hence much of the machinery involved in (Koh & Liang (2017)) can be imported into our framework. With $p$ parameters in $\theta$, inverting $H_\theta$ requires $O(np^2 + p^3)$ operations, which can be cumbersome for deep learning models with large number of parameters. Following (Koh & Liang (2017)), we avoid explicitly computing $H_\theta$, and instead use implicit Hessian-vector products to efficiently approximate the Hessian products in (9). For models with a very large $p$, we can use the stochastic estimation approach in (Agarwal et al. (2016)) to estimate $H_\theta^{-1}$ with $O(p)$ complexity.

### 3.4 Theoretical Guarantees

Now that we have developed a practical algorithm for implementing the DJ, we conclude this Section by revisiting the performance requirements in Section 2. In the following Theorem, we show that the approximate DJ confidence intervals satisfy the desired coverage and discrimination requirements.

**Theorem 2.** *Let $\widehat{\mathcal{C}}_{\alpha,n}^{(m)}(\boldsymbol{x}; \hat{\theta})$ be the approximate DJ confidence interval evaluated with $m$ HOIFs using Algorithm 1. Then, for any model $f(\boldsymbol{x}; \hat{\theta})$, the coverage rate achieved by the DJ is*

$$\lim_{m \to \infty} \mathbb{P} \Big\{ y \in \widehat{\mathcal{C}}_{\alpha,n}^{(m)}(\boldsymbol{x}; \hat{\theta}) \Big\} \geq 1 - \alpha.$$

*If the model is stable, i.e., $\mathbb{P}\{ |f(\boldsymbol{x}; \hat{\theta}) - f(\boldsymbol{x}; \hat{\theta}_{-i})| \leq \varepsilon \} \geq 1 - \delta$, $\forall i$, for some $\varepsilon$ and $\delta \in (0, 1)$ (Bousquet & Elisseeff (2002)), then $\forall \boldsymbol{x}, \boldsymbol{x}' \in \mathcal{X}$, $y, y' \in \mathcal{Y}$, and for a sufficiently large $n$, we have*

$$\mathbb{E}[\, W(\widehat{\mathcal{C}}_{\alpha,n}^{(\infty)}(\boldsymbol{x}; \hat{\theta}))\,] \geq \mathbb{E}[\, W(\widehat{\mathcal{C}}_{\alpha,n}^{(\infty)}(\boldsymbol{x}'; \hat{\theta}))\,] \Leftrightarrow \mathbb{E}[\, \ell(y, f(\boldsymbol{x}; \hat{\theta}))\,] \geq \mathbb{E}[\, \ell(y', f(\boldsymbol{x}'; \hat{\theta}))\,]. \quad \square$$

Theorem 2 provides a very strong guarantee on the frequentist coverage of the DJ confidence intervals. That is, without any assumptions, the exact DJ interval $\widehat{\mathcal{C}}_{\alpha,n}^{(\infty)}(\boldsymbol{x}; \hat{\theta})$ has a worst case coverage that fulfills the target coverage rate of $(1 - \alpha)$. (The approximate DJ intervals, $\widehat{\mathcal{C}}_{\alpha,n}^{(m)}(\boldsymbol{x}; \hat{\theta})$, achieve the target coverage in the limit of $m \to \infty$.) In Section 5, we show through empirical evaluation that in practice — even with a finite number of HOIFs — the DJ intervals will achieve the $(1 - \alpha)$ coverage. With further assumption on algorithmic stability (Bousquet & Elisseeff (2002)), Theorem 2 states that if a large number of HOIFs are involved in constructing the approximate DJ intervals, then the discrimination condition will be satisfied if we have a sufficiently large training sample.

---

[3]All proofs are provided in the supplementary appendix.

## 4 RELATED WORK

The DJ is a *post-hoc* method that operates on deep learning models after being trained. Post-hoc methods are advantageous in that they neither compromise the model accuracy, nor require any modifications in model structure or training. (To implement Algorithm 1 (Section 3.3), we only need access to the model's loss gradients and Hessian-vector products.) It is worth noting that, while our primary motivation is to estimate uncertainty in deep learning models, our method is applicable to any machine learning model for which HOIFs are accessible. Additionally, the exact DJ procedure can be applied to any model via exhaustive leave-one-out re-training, even when gradients are inaccessible.

Post-hoc methods have been traditionally underexplored since existing approaches, such as calibration via temperature scaling (Platt et al. (1999)), are known to under-perform compared to built-in methods (Ovadia et al. (2019)). However, recent works have revived post-hoc approaches using ideas based on bootstrapping (Schulam & Saria (2019)), jackknife resampling (Barber et al. (2019b); Giordano et al. (2018)) and cross-validation (Vovk et al. (2018); Barber et al. (2019a)). An overview of the different classes of post-hoc and built-in methods proposed in recent literature is provided in Table 1.

Table 1: Overview of existing methods for uncertainty quantification in deep learning.

| Method | Bayesian vs. Frequentist | Post-hoc vs. Built-in | Coverage |
|---|---|---|---|
| Bayesian neural nets (Ritter et al. (2018)) | Bayesian | Built-in | No guarantees |
| Probabilistic backprop. (Blundell et al. (2015)) | Bayesian | Built-in | No guarantees |
| Monte Carlo dropout (Gal & Ghahramani (2016)) | Bayesian | Built-in | No guarantees |
| Deep ensembles (Lakshminarayanan et al. (2017)) | Frequentist | Built-in | No guarantees |
| Resampling uncertainty (Schulam & Saria (2019)) | Frequentist | Post-hoc | No guarantees |
| Cross-conformal (Vovk et al. (2018)) | Frequentist | Post-hoc | $1 - 2\alpha$ |
| Jackknife+ (Barber et al. (2019b)) | Frequentist | Post-hoc | $1 - 2\alpha$ |
| Jackknife-minmax (Barber et al. (2019b)) | Frequentist | Post-hoc | $1 - \alpha$ |

As we discussed earlier, Bayesianism has been the dominant approach to uncertainty quantification in deep learning (Welling & Teh (2011); Hernández-Lobato & Adams (2015); Ritter et al. (2018); Maddox et al. (2019)). By their very nature, Bayesian models cannot be applied in a post-hoc fashion since they require specifying priors over model parameters, which leads to major modifications in the inference algorithms. While Bayesian models provide a formal framework for uncertainty estimation, posterior credible intervals do not guarantee frequentist coverage, and more crucially, the achieved coverage can be very sensitive to hyper-parameter tuning (Bayarri & Berger (2004)). Moreover, exact Bayesian inference is often computationally prohibitive, and alternative approximations — e.g., (Gal & Ghahramani (2016)) and (Kingma et al. (2015))) — may induced posterior distributions that do not concentrate asymptotically (Osband (2016); Hron et al. (2017)).

Deep ensembles (Lakshminarayanan et al. (2017)) are currently considered the most competitive (non-Bayesian) benchmark for uncertainty quantification (Ovadia et al. (2019)). This method repeatedly re-trains the model on sub-samples of the data (using adversarial training), and then estimates uncertainty through the variance of the aggregate predictions. A similar bootstrapping approach, developed in (Schulam & Saria (2019)), uses the model's Hessian and loss gradients to create an ensemble without re-training. While these methods may perform favorably in terms of discrimination, they are likely to undercover, since they only consider local variability terms akin to those in (5). Additionally, ensemble methods do not provide theoretical guarantees on coverage and discrimination.

The (infinitesimal) jackknife method was previously applied for quantifying the predictive uncertainty in random forest models (Wager et al. (2014); Mentch & Hooker (2016); Wager & Athey (2018)). In these works, however, the developed jackknife estimators are bespokely tailored to bagging predictors, and cannot be straightforwardly extended to deep neural networks. More recently, general-purpose jackknife estimators were developed in (Barber et al. (2019b)), where two exhaustive leave-one-out procedures: the *jackknife+* and the *jackknife-minmax* where shown to have assumption-free worst-case coverage guarantees of $(1 - 2\alpha)$ and $(1 - \alpha)$, respectively. Our work improves on these results in two ways. First, the DJ achieves the target coverage rate of $(1 - \alpha)$ with a strictly narrower interval width compared to the jackknife-minmax (proof is provided in Appendix D). Second, our proposed HOIF-based implementation of the jackknife alleviates the need for exhaustive leave-one-out re-training, which enables scaling up *any* jackknife-based method to large datasets.

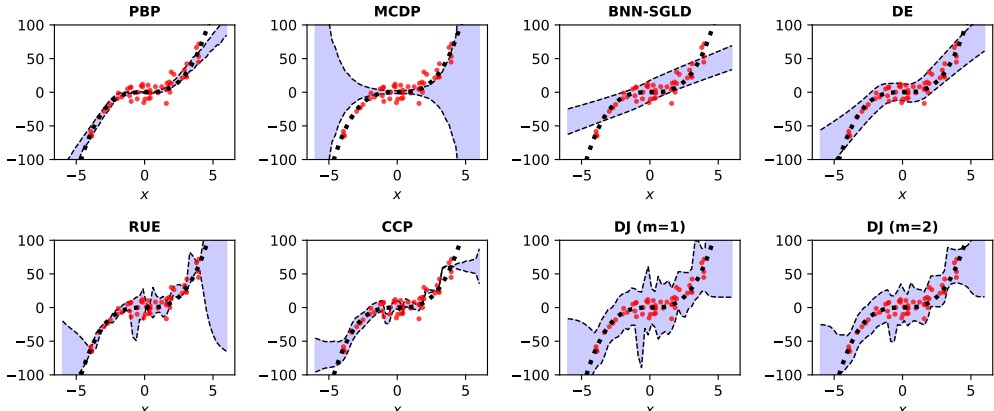

Figure 3: Uncertainty estimates by all baselines on the synthetic dataset. Training data is shown as red dots and the true function is displayed as a black dotted line. Confidence intervals are shown as a shaded blue region.

## 5 EXPERIMENTS

Our DJ procedure (Algorithm 1) was implemented on top of `PyTorch` models, where we computed the loss gradients and Hessian required for evaluating the HOIFs of a model's parameters (Theorem 1) using `autograd` (Maclaurin et al. (2015)). The only hyper-parameter involved in our method is the number of HOIFs $m$ — this was tuned by optimizing the evaluation metrics (listed below) on the training samples. Code snippets for the DJ procedure are provided in the supplementary appendix.

**Baselines.** We compared our DJ procedure with 6 state-of-the-art baseline methods. These included 3 built-in Bayesian methods: (1) Monte Carlo Dropout (MCDP) (Gal & Ghahramani (2016)), (2) Probabilistic backpropagation (PBP) (Hernández-Lobato & Adams (2015)), and (3) Bayesian neural networks with inference via stochastic gradient Langevin dynamics (BNN-SGLD) (Welling & Teh (2011)). In addition, we considered deep ensembles (DE) (Lakshminarayanan et al. (2017)), which is deemed the most competitive built-in frequentist method (Ovadia et al. (2019)). We also included 2 post-hoc methods: resampling uncertainty estimation (RUE) (Schulam & Saria (2019)) and cross-conformal predictive distributions (CCP) (Vovk et al. (2018)). For a target coverage of $(1 - \alpha)$, uncertainty estimates were obtained by setting the posterior quantile functions to $(1-\alpha)$ (for Bayesian methods), or obtaining the $(1 - \alpha)$ percentile point of a normal distribution (for frequentist methods). We implemented all baselines in `PyTorch` (details are provided in the supplementary appendix).

**Evaluation metrics.** In all experiments, we used the mean squared error (MSE) as the loss $L(\mathcal{D}, \hat{\theta})$ for the trained model $f(\boldsymbol{x}; \theta)$. To ensure a fair comparison, the hyper-parameters of the model $f(\boldsymbol{x}; \theta)$ were the same for all baselines. In each experiment, an uncertainty estimate $\widehat{\mathcal{C}}_{\alpha,n}(\boldsymbol{x}; \theta)$ is obtained from a training sample, and then coverage and discrimination are evaluated on a test sample. To evaluate empirical coverage probability, we compute the fraction of test samples for which $y$ resides in $\widehat{\mathcal{C}}_{\alpha,n}(\boldsymbol{x}; \hat{\theta})$. Discrimination was evaluated as follows. For each baseline method, we evaluate the interval width $W(\widehat{\mathcal{C}}_{\alpha,n}(\boldsymbol{x}; \theta))$ for all test points. For a given error threshold $\mathcal{E}$, we use the interval width to detect whether the test prediction $f(\boldsymbol{x}; \theta)$ is a high-confidence, i.e., $\ell(y, f(\boldsymbol{x}; \theta)) \leq \mathcal{E}$, or a low-confidence prediction, i.e., $\ell(y, f(\boldsymbol{x}; \theta)) > \mathcal{E}$. We use the area under the ROC curve (AUC-ROC) to evaluate the accuracy of classifying high- and low-confidence predictions using $W(\widehat{\mathcal{C}}_{\alpha,n}(\boldsymbol{x}; \theta))$.

### 5.1 SYNTHETIC DATA

**Qualitative assessment.** First, we qualitatively assess the DJ confidence intervals using the synthetic data model in (Hernández-Lobato & Adams (2015)). This dataset is based on the one-dimensional model: $y = x^3 + \epsilon$, where $\epsilon \sim \mathcal{N}(0, 3^2)$. We fit a feed-forward neural network model (with the same architecture as (Hernández-Lobato & Adams (2015))) on 50 training examples drawn from a uniform distribution on $\mathcal{X} = [-4, 4]$, and evaluate confidence intervals on its predictions with a target coverage of 90%. The confidence intervals constructed by all baselines are visualized in Figure 3.

As we can see in Figure 3, Bayesian methods tend to undercover the target function — i.e., PBP overestimated model confidence, whereas BNN-SGLD failed to recognize that the region $|x| > 4$ should be assigned higher uncertainty since it contains no training data. For MCDP, we can see that higher uncertainty is assigned to the region $|x| > 4$, but the posterior does not concentrate around the training data, and hence credible intervals are poorly calibrated. Post-hoc methods (RUE and CCP) tend to overestimate confidence in regions with dense training data ($|x| \leq 4$) since they rely mainly on the (in-sample) error variance to estimate uncertainty. DE produces a favorable confidence interval, but with little variation in interval width and with few training points being uncovered.

The DJ confidence intervals hit the right balance between coverage and discrimination. The addition of the marginal error $\widehat{Q}_{0.1,50}(\mathcal{R}_{50})$ in (5) ensures that the training data points and target function are covered everywhere in $\mathcal{X}$. The variability terms, $\widehat{Q}_{0.1,23}(\mathcal{V}_{50}^+)$ and $\widehat{Q}_{0.9,27}(\mathcal{V}_{50}^-)$, add the discrimination components, which enable the interval width to clearly distinguish high-confidence regions ($|x| \leq 4$) from low-confidence ones ($|x| > 4$). By contrasting the DJ estimates with $m = 1$ and $m = 2$, we see that adding more HOIF terms gives smoother and tighter confidence intervals.

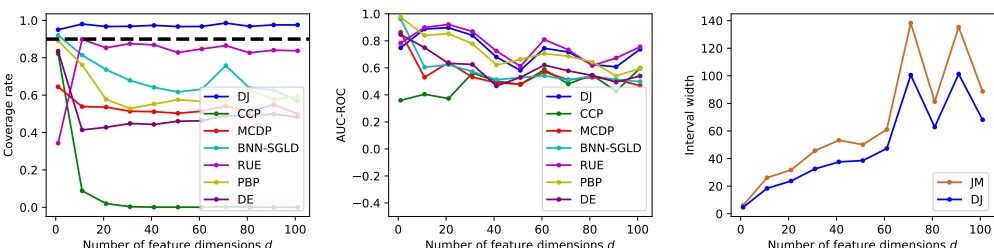

Figure 4: Coverage and discrimination performance of all baselines on high-dimensional synthetic data.

**Quantitative assessment.** Next, we examine coverage and discrimination performance of all baselines on the following synthetic model. For $n = 100$ and $1 - \alpha = 0.9$, we conduct 100 experiments, each for different feature dimensions $d = 1, 11, 21, \ldots, 101$, with i.i.d. samples $(\boldsymbol{x}_i, y_i)$ generated as

$$\boldsymbol{\beta} \sim Uniform[0, 1]^{\otimes d}, \ \boldsymbol{x}_i \sim \mathcal{N}(0, \boldsymbol{I}_d), \ \text{and} \ y_i \,|\, \boldsymbol{x}_i \sim \mathcal{N}((\boldsymbol{x}_i^T \boldsymbol{\beta})^3, 1).$$

In each experiment, we fit a 2-layer neural network with 100 hidden units to the training data, and then generate 100 test data points from the same distribution to evaluate the coverage and discrimination performance for different values of $d$. (Discrimination is evaluated by setting the error threshold $\mathcal{E}$ to be the $95\%$ percentile of the trained model error on the test samples.) As we can see in Figure 4, DJ is the only method that always manages to achieve the $90\%$ coverage rate for any number of features $d$ (as predicted by Theorem 2). It does so while also achieving the most competitive discrimination (AUC-ROC) performance, together with the RUE and PBP baselines. We also found that, for all values of $d$, the DJ interval width was strictly tighter than that of the jackknife-minmax (JM) procedure (Barber et al. (2019b)), which is the only other procedure that guarantees the $1 - \alpha$ target coverage.

In this experiment, we found that $m = 2$ HOIFs are sufficient for achieving the best possible performance by DJ. More HOIFs would generally be needed for more complex networks, with more layers or hidden units (see further results in the supplementary appendix). Because of its post-hoc nature, the computational complexity of DJ was significantly less than built-in method (average runtime of DJ was 50 times faster than DE, 30 times faster than BNN-SGLD and 10 times faster than PBP).

## 5.2 REAL DATA: AUDITING MODEL RELIABILITY

We conducted a series of experiments on real-world datasets in order to evaluate the accuracy of uncertainty estimates issued by the DJ procedure in various contexts. In this Section, we show how uncertainty estimates can be used to audit the reliability of a predictive model. Further experiments on *active learning* and *out-of-distribution detection* are deferred to the supplementary appendix.

In this experiment, we compare how well DJ and the competing baselines are able to detect when a model's prediction will be far from the true target value. We conduct our experiments on 8 common UCI benchmark datasets for regression — those included: Boston housing, concrete compression strength, energy efficiency, robot kinematics (Kin8nm), naval propulsion, combined cycle power plant, red wine quality, and yacht hydrodynamics (Dua & Graff (2017)).

| | Dataset | | | | | | | |
|---|---|---|---|---|---|---|---|---|
| **Method** | **Housing** | **Concrete** | **Energy** | **Kin8nm** | **Naval** | **Power** | **Wine** | **Yacht** |
| **DJ** | $\mathbf{0.93 \pm 0.01}$ | $0.55 \pm 0.01$ | $\mathbf{0.92 \pm 0.01}$ | $\mathbf{0.71 \pm 0.01}$ | $\mathbf{0.89 \pm 0.01}$ | $\mathbf{0.58 \pm 0.02}$ | $0.69 \pm 0.01$ | $\mathbf{0.92 \pm 0.01}$ |
| ($m = 10$) | $(92.61\%)^*$ | $(96.12\%)^*$ | $(99.40\%)^*$ | $(95.58\%)^*$ | $(98.91\%)^*$ | $(95.36\%)^*$ | $(94.87\%)^*$ | $(100\%)^*$ |
| **DJ** | $0.90 \pm 0.01$ | $0.54 \pm 0.012$ | $\mathbf{0.92 \pm 0.01}$ | $0.66 \pm 0.01$ | $0.85 \pm 0.01$ | $0.53 \pm 0.01$ | $0.69 \pm 0.01$ | $0.90 \pm 0.01$ |
| ($m = 1$) | $(93.52\%)^*$ | $(97.22\%)^*$ | $(94.66\%)^*$ | $(91.32\%)^*$ | $(97.22\%)^*$ | $(95.21\%)^*$ | $(92.71\%)^*$ | $(100\%)^*$ |
| **RUE** | $0.88 \pm 0.02$ | $0.51 \pm 0.01$ | $0.90 \pm 0.01$ | $\mathbf{0.71 \pm 0.01}$ | $\mathbf{0.89 \pm 0.02}$ | $0.52 \pm 0.03$ | $0.69 \pm 0.02$ | $0.90 \pm 0.01$ |
| | $(64.77\%)$ | $(78.17\%)$ | $(97.30\%)^*$ | $(61.08\%)$ | $(52.21\%)$ | $(82.85\%)$ | $(80.76\%)$ | $(100\%)^*$ |
| **MCDP** | $0.70 \pm 0.01$ | $0.51 \pm 0.01$ | $0.62 \pm 0.03$ | $0.58 \pm 0.00$ | $0.62 \pm 0.01$ | $0.51 \pm 0.04$ | $0.50 \pm 0.00$ | $0.76 \pm 0.01$ |
| | $(85.55\%)$ | $(55.15\%)$ | $(0.0\%)$ | $(81.15\%)$ | $(65.22\%)$ | $(50.15\%)$ | $(0.0\%)$ | $(0\%)$ |
| **PBP** | $0.51 \pm 0.01$ | $0.55 \pm 0.01$ | $0.51 \pm 0.03$ | $0.58 \pm 0.03$ | $0.51 \pm 0.01$ | $0.50 \pm 0.02$ | $\mathbf{0.80 \pm 0.03}$ | $0.89 \pm 0.02$ |
| | $(8.86\%)$ | $(2.58\%)$ | $(1.49\%)$ | $(99.78\%)^*$ | $(97.87\%)^*$ | $(40.23\%)$ | $(12.82\%)$ | $(89.98\%)$ |
| **DE** | $0.89 \pm 0.02$ | $0.77 \pm 0.01$ | $0.88 \pm 0.03$ | $0.58 \pm 0.02$ | $0.68 \pm 0.02$ | $0.50 \pm 0.01$ | $0.55 \pm 0.01$ | $0.52 \pm 0.03$ |
| | $(84.48\%)$ | $(77.52\%)$ | $(76.79\%)$ | $(54.77\%)$ | $(81.48\%)$ | $(97.36\%)^*$ | $(42.18\%)$ | $(52.98\%)$ |
| **BNN** | $0.80 \pm 0.01$ | $\mathbf{0.79 \pm 0.02}$ | $0.62 \pm 0.01$ | $0.63 \pm 0.00$ | $0.55 \pm 0.01$ | $0.50 \pm 0.01$ | $0.50 \pm 0.00$ | $0.59 \pm 0.01$ |
| | $(88.67\%)$ | $(100\%)^*$ | $(100\%)^*$ | $(94.21\%)^*$ | $(64.21\%)$ | $(62.13\%)$ | $(84.43\%)$ | $(83.44\%)$ |
| **CCP** | $0.51 \pm 0.01$ | $0.57 \pm 0.01$ | $0.63 \pm 0.01$ | $0.52 \pm 0.01$ | $0.50 \pm 0.01$ | $0.50 \pm 0.01$ | $0.57 \pm 0.02$ | $0.74 \pm 0.01$ |
| | $(87.43\%)$ | $(78.17\%)$ | $(85.32\%)$ | $(86.61\%)$ | $(87.72\%)$ | $(87.67\%)$ | $(88.46\%)$ | $(80.79\%)$ |

Table 2: AUC-ROC performance ($\pm$ 95% confidence intervals) of all baselines on the real-world UCI regression datasets. Numbers between brackets are the achieved (empirical) coverage rates. Bold numbers correspond to the best performing method. Coverage rates marked with an asterisk achieve the 90% target rate.

For each dataset, we sample 20 train-test splits, and use 90% of the data points for training. To control for model complexity, we fix a single feed-forward neural network architecture with two hidden layers and 100 hidden units, logistic activation functions, MSE loss, and a single set of learning hyper-parameters (500 epochs with 100 samples per minibatch, and an Adam optimizer with default settings). We fit a single model for each split of each dataset, and compute the confidence intervals (with $1 - \alpha = 0.9$) for all baselines on the test data. We implemented two versions of our method: a DJ with a first-order von Mises approximation ($m = 1$), and a DJ with $m = 10$ HOIFs. Similar to the previous experiment, we evaluate the empirical coverage rate and discrimination (AUC-ROC) performance assuming the threshold $\mathcal{E}$ to be the 95% percentile of the empirical distribution over test errors. Results are provided in Table 2.

We observe that, in terms of discriminative performance, DJ outperforms (or is competitive with) the baselines — the AUC-ROC of detecting low-quality predictions by DJ was better than other methods in 6 out of the 8 datasets. Including HOIFs was important for achieving good performance in most datasets. A first-order von Mises approximation was sufficient only in the "Energy" dataset, but under-performed compared to the DJ with $m = 10$ in all other datasets. Another key observation is that — as predicted by Theorem 2 — the DJ procedure always achieves the 90% target coverage rate. All other methods tend to undercover the test points, and some methods (such as MCDP and PBP) can even achieve close to zero coverage. It is worth mentioning that on the dataset where DJ provided the biggest improvement in discriminative performance (Boston housing), it was also the only method that achieved the target coverage.

## 6 CONCLUSION

Uncertainty quantification is a crucial requirement in various high-stakes applications where deep learning can inform critical decision-making. In this paper, we developed a rigorous frequentist procedure for quantifying the uncertainty in predictions issued by deep learning models in a post-hoc fashion. Our procedure, which is inspired by classical jackknife estimators, does not require any modifications in the underlying deep learning model, and provides theoretical guarantees on its achieved performance. Implementation of our procedure using higher-order influence functions entails minimal computational costs, and allows our method to scale for big datasets, without the need for repeated re-training as in the classical jackknife procedure. While this paper focuses on uncertainty estimation in deep learning, the methods therein can be applied to any machine learning model as long as we are granted access to its loss gradients and Hessian.

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

## A   INFLUENCE FUNCTIONS: BACKGROUND & KEY CONCEPTS

### A.1   FORMAL DEFINITION

Robust statistics is the branch of statistics concerned with the detection of outlying observations. An estimator is deemed *robust* if it produces similar results as the majority of observations indicates, regardless of how a minority of other observations is perturbed (Huber & Ronchetti (1981)). The influence function measures these effects in statistical functionals by analyzing the behavior of a functional not only at the distribution of interest, but also in an entire neighborhood of distributions around it. Lack of model robustness is a clear indicator of model uncertainty, and hence influence functions arise naturally in our method as a (pointwise) surrogate measure of model uncertainty. In this section we formally define influence functions and discuss its properties.

The pioneering works in (Hampel et al. (2011)) and (Huber & Ronchetti (1981)) coined the notion of influence functions to assess the robustness of statistical functionals to perturbations in the underlying distributions. Consider a statistical functional $T : \mathcal{P} \to \mathbb{R}$, defined on a probability space $\mathcal{P}$, and a probability distribution $\mathbb{P} \in \mathcal{P}$. Consider distributions of the form $\mathbb{P}_{\varepsilon,z} = (1 - \varepsilon)\mathbb{P} + \varepsilon\Delta z$ where $\Delta z$ denotes the Dirac distribution in the point $z = (\boldsymbol{x}, y)$, representing the contaminated part of the data. For the functional $T$ to be considered robust, $T(\mathbb{P}_{\varepsilon,z})$ should not be too far away from $T(\mathbb{P})$ for any possible $z$ and any small $\varepsilon$. The limiting case of $\varepsilon \to 0$ defines the influence function. That is, Then the influence function of $T$ at $\mathbb{P}$ in the point $z$ is defined as

$$\mathcal{I}(z; \mathbb{P}) = \lim_{\varepsilon \to 0} \frac{T(\mathbb{P}_{\varepsilon,z}) - T(\mathbb{P})}{\varepsilon} \triangleq \frac{\partial}{\partial \varepsilon} T(\mathbb{P}_{\varepsilon,z}) \bigg|_{\varepsilon=0}, \qquad (A.1)$$

The influence function measures the robustness of $T$ by quantifying the effect on the estimator $T$ when adding an infinitesimally small amount of contamination at the point $z$. If the supremum of $\mathcal{I}(.)$ over $z$ is bounded, then an infinitesimally small amount of perturbation cannot cause arbitrary large changes in the estimate. Then small amounts of perturbation cannot completely change the estimate which ensures the robustness of the estimator.

### A.2   THE VON MISES EXPANSION

The von Mises expansion is a distributional analog of the Taylor expansion applied for a functional instead of a function. For two distributions $\mathbb{P}$ and $\mathbb{Q}$, the Von Mises expansion is (Fernholz (2012)):

$$T(\mathbb{Q}) = T(\mathbb{P}) + \int \mathcal{I}^{(1)}(z; \mathbb{P}) \, d(\mathbb{Q} - \mathbb{P}) + \frac{1}{2} \int \mathcal{I}^{(2)}(z; \mathbb{P}) \, d(\mathbb{Q} - \mathbb{P}) + \dots, \qquad (A.2)$$

where $\mathcal{I}^{(k)}(z; \mathbb{P})$ is the $k^{th}$ order influence function. By setting $\mathbb{Q}$ to be a perturbed version of $\mathbb{P}$, i.e., $\mathbb{Q} = \mathbb{P}_\varepsilon$, the von Mises expansion at point $z$ reduces to:

$$T(\mathbb{P}_{\varepsilon,z}) = T(\mathbb{P}) + \varepsilon \, \mathcal{I}^{(1)}(z; \mathbb{P}) + \frac{\varepsilon^2}{2} \, \mathcal{I}^{(2)}(z; \mathbb{P}) + \dots, \qquad (A.3)$$

and so the $k^{th}$ order influence function is operationalized through the derivative

$$\mathcal{I}^{(k)}(z; \mathbb{P}) \triangleq \frac{\partial}{\partial^k \varepsilon} T(\mathbb{P}_{\varepsilon,z}) \bigg|_{\varepsilon=0}. \qquad (A.4)$$

### A.3   INFLUENCE FUNCTION OF MODEL LOSS

Now we apply the mathematical definitions in Sections A.1 and A.2 to our learning setup. In our setting, the functional $T(.)$ corresponds to the (trained) model parameters $\hat{\theta}$ and the distribution $\mathbb{P}$. In this case, influence functions of $\hat{\theta}$ computes how much the model parameters would change if the underlying data distribution was perturbed infinitesimally.

$$\mathcal{I}_{\theta}^{(1)}(z) = \frac{\partial \hat{\theta}_{\varepsilon,z}}{\partial \varepsilon} \bigg|_{\varepsilon=0}, \quad \hat{\theta}_{\varepsilon,z} \triangleq \arg\min_{\theta \in \Theta} \frac{1}{n} \sum_{i=1}^{n} \ell(z_i; \theta) + \varepsilon \, \ell(z; \theta). \qquad (A.5)$$

Recall that in the definition of the influence function $\mathbb{P}_{\varepsilon,z} = (1 - \varepsilon)\mathbb{P} + \varepsilon\Delta z$ where $\Delta z$ denotes the Dirac distribution in the point $z = (\boldsymbol{x}, y)$. Thus, the (first-order) influence function in (A.5)

corresponds to perturbing a training data point $z$ by an infinitesimally small change $\varepsilon$ and evaluating the corresponding change in the learned model parameters $\hat{\theta}$. More generally, the $k^{th}$ order influence function of $\hat{\theta}$ is defined as follows:

$$\mathcal{I}_{\theta}^{(k)}(z) = \frac{\partial^k \hat{\theta}_{\varepsilon,z}}{\partial \varepsilon^k}\bigg|_{\varepsilon=0}. \tag{A.6}$$

By applying the von Mises expansion, we can approximate the parameter of a model trained on the training dataset with perturbed data point $z$ as follows:

$$\hat{\theta}_{\varepsilon,z} \approx \hat{\theta} + \varepsilon \mathcal{I}_{\theta}^{(1)}(z) + \frac{\varepsilon^2}{2} \mathcal{I}_{\theta}^{(2)}(z) + \ldots + \frac{\varepsilon^m}{m!} \mathcal{I}_{\theta}^{(m)}(z), \tag{A.7}$$

where $m$ is the number of terms included in the truncated expansion. When $m = \infty$, the exact parameter $\hat{\theta}_{\varepsilon,z}$ without the need to re-train the model.

### A.4 CONNECTION TO LEAVE-ONE-OUT ESTIMATORS

Our uncertainty estimator depends on perturbing the model parameters by removing a single training point at a time. Note that removing a point $z$ is the same as perturbing $z$ by $\varepsilon = \frac{-1}{n}$, hence we obtain an ($m^{th}$ order) approximation of the parameter change due to removing the point $z$ as follows:

$$\hat{\theta}_{-z} - \hat{\theta} \approx \frac{-1}{n} \mathcal{I}_{\theta}^{(1)}(z) + \frac{1}{2n^2} \mathcal{I}_{\theta}^{(2)}(z) + \ldots + \frac{(-1)^m}{n^m \cdot m!} \mathcal{I}_{\theta}^{(m)}(z), \tag{A.8}$$

where $\hat{\theta}_{-z}$ is the model parameter learned by removing the data point $z$ from the training data.

## B PROOF OF THEOREM 1

Recall that the LOO parameter $\hat{\theta}_{i,\epsilon}$ is obtained by solving the optimization problem:

$$\hat{\theta}_{i,\epsilon} = \arg\min_{\theta \in \Theta} L(\mathcal{D}, \theta) + \epsilon \cdot \ell(y_i, f(\boldsymbol{x}_i; \theta)). \tag{B.1}$$

Let us first derive the first order influence function $\mathcal{I}^{(1)}(\boldsymbol{x}_i, y_i)$. Let us first define $\Delta_{i,\epsilon} \triangleq \hat{\theta}_{i,\epsilon} - \hat{\theta}$. The first order influence function is given by:

$$\mathcal{I}^{(1)}(\boldsymbol{x}_i, y_i) = \frac{\partial \hat{\theta}_{i,\epsilon}}{\partial \epsilon} = \frac{\partial \Delta_{i,\epsilon}}{\partial \epsilon}. \tag{B.2}$$

Note that, since $\hat{\theta}_{i,\epsilon}$ is the minimizer of (B.1), then the perturbed loss has to satisfy the following (first order) optimality condition:

$$\nabla_\theta \left\{ L(\mathcal{D}, \theta) + \epsilon \cdot \ell(y_i, f(\boldsymbol{x}_i; \theta)) \right\}\big|_{\theta = \hat{\theta}_{i,\epsilon}} = 0. \tag{B.3}$$

Since $\lim_{\epsilon \to 0} \hat{\theta}_{i,\epsilon} = \hat{\theta}$, then we can write the following Taylor expansion:

$$\nabla_\theta \sum_{k=0}^{\infty} \frac{\Delta_{i,\epsilon}^k}{k!} \cdot \nabla_\theta^k \left\{ L(\mathcal{D}, \hat{\theta}) + \epsilon \cdot \ell(y_i, f(\boldsymbol{x}_i; \hat{\theta})) \right\} = 0. \tag{B.4}$$

Now by dropping the $o(\|\Delta_{i,\epsilon}\|)$ terms, we have:

$$\nabla_\theta \left( \left\{ L(\mathcal{D}, \hat{\theta}) + \epsilon \cdot \ell(y_i, f(\boldsymbol{x}_i; \hat{\theta})) \right\} + \Delta_{i,\epsilon} \cdot \nabla_\theta \left\{ L(\mathcal{D}, \hat{\theta}) + \epsilon \cdot \ell(y_i, f(\boldsymbol{x}_i; \hat{\theta})) \right\} \right) = 0. \tag{B.5}$$

Since $\hat{\theta}$ is a indeed a minimizer of the loss function $\ell(.)$, then we have $\nabla_\theta \ell(.) = 0$. Thus, (B.5) reduces to the following condition:

$$\left\{ \epsilon \cdot \nabla_\theta \ell(y_i, f(\boldsymbol{x}_i; \hat{\theta})) \right\} + \Delta_{i,\epsilon} \cdot \left\{ \nabla_\theta^2 L(\mathcal{D}, \hat{\theta}) + \epsilon \cdot \nabla_\theta^2 \ell(y_i, f(\boldsymbol{x}_i; \hat{\theta})) \right\} = 0. \tag{B.6}$$

By solving for $\nabla_\theta$, we have

$$\Delta_{i,\epsilon} = - \left\{ \nabla_\theta^2 L(\mathcal{D}, \hat{\theta}) + \epsilon \cdot \nabla_\theta^2 \ell(y_i, f(\boldsymbol{x}_i; \hat{\theta})) \right\}^{-1} \cdot \left\{ \epsilon \cdot \nabla_\theta \ell(y_i, f(\boldsymbol{x}_i; \hat{\theta})) \right\}, \tag{B.7}$$

which can be approximated by keeping only the $O(\epsilon)$ terms as follows:

$$\Delta_{i,\epsilon} = -\left\{\nabla_\theta^2 L(\mathcal{D},\hat{\theta})\right\}^{-1} \cdot \left\{\epsilon \cdot \nabla_\theta \ell(y_i, f(\boldsymbol{x}_i;\hat{\theta}))\right\}. \tag{B.8}$$

Noting that $\nabla_\theta^2 L(\mathcal{D},\hat{\theta})$ is the Hessian matrix $H_{\hat{\theta}}$, we have:

$$\Delta_{i,\epsilon} = -H_{\hat{\theta}}^{-1} \cdot \left\{\epsilon \cdot \nabla_\theta \ell(y_i, f(\boldsymbol{x}_i;\hat{\theta}))\right\}. \tag{B.9}$$

By taking the derivative with respect to $\epsilon$, we arrive at the expression for first order influence functions:

$$\mathcal{I}^{(1)}(\boldsymbol{x}_i, y_i) = \left.\frac{\Delta_{i,\epsilon}}{\epsilon}\right|_{\epsilon=0} = -H_{\hat{\theta}}^{-1} \cdot \nabla_\theta \ell(y_i, f(\boldsymbol{x}_i;\hat{\theta})). \tag{B.10}$$

Now let us examine the second order influence functions. In order to obtain $\mathcal{I}^{(2)}(\boldsymbol{x}_i, y_i)$, we need to differentiate (B.6) after omitting the $O(\epsilon)$ once again as follows:

$$\left\{2\Delta_{i,\epsilon} \cdot \epsilon \cdot \nabla_\theta^2 \ell(y_i, f(\boldsymbol{x}_i;\hat{\theta}))\right\} + \left\{\Delta_{i,\epsilon}^2 \cdot \nabla_\theta^2 L(\mathcal{D},\hat{\theta}) + \Delta_{i,\epsilon} \cdot \Delta_{i,\epsilon} \cdot \nabla_\theta^3 L(\mathcal{D},\hat{\theta})\right\} = 0. \tag{B.11}$$

Where we have applied the chain rule to obtain the above. By substituting $\nabla_\theta^2 L(\mathcal{D},\hat{\theta}) = H_\theta$ and dividing both sides of (B.11) by $\epsilon^2$, we have

$$\left\{2\frac{\Delta_{i,\epsilon}}{\epsilon} \cdot \nabla_\theta^2 \ell(y_i, f(\boldsymbol{x}_i;\hat{\theta}))\right\} + \left\{\frac{\Delta_{i,\epsilon}^2}{\epsilon^2} \cdot H_\theta + \left(\frac{\Delta_{i,\epsilon}}{\epsilon}\right)^2 \cdot \nabla_\theta^3 L(\mathcal{D},\hat{\theta})\right\} = 0. \tag{B.12}$$

Thus, by re-arranging (B.11), we can obtain $\mathcal{I}^{(2)}(\boldsymbol{x}_i, y_i)$ in terms of $\mathcal{I}^{(1)}(\boldsymbol{x}_i, y_i)$ as follows:

$$\mathcal{I}^{(2)}(\boldsymbol{x}_i, y_i) = -H_\theta^{-1}\left(\left(\mathcal{I}^{(1)}(\boldsymbol{x}_i, y_i)\right)^2 \cdot \nabla_\theta^3 L(\mathcal{D},\hat{\theta}) + 2\,\mathcal{I}^{(1)}(\boldsymbol{x}_i, y_i) \cdot \nabla_\theta^2 \ell(y_i, f(\boldsymbol{x}_i;\hat{\theta}))\right).$$

Similarly, we can obtain the $k^{th}$ order influence function, for any $k > 1$, by repeatedly differentiating equation (B.6) $k$ times, i.e.,

$$\frac{\partial}{\partial \epsilon^k}\left\{\epsilon \cdot \nabla_\theta \ell(y_i, f(\boldsymbol{x}_i;\hat{\theta})) + \Delta_{i,\epsilon} \cdot \nabla_\theta^2 L(\mathcal{D},\hat{\theta})\right\} = 0. \tag{B.13}$$

and solving for $\partial \Delta_{i,\epsilon}^k/\partial \epsilon^k$. By applying the higher-order chain rule to the two added terms in (B.13) (or equivalently, take the derivative of $\mathcal{I}^{(2)}(\boldsymbol{x}_i, y_i)$ for $k - 2$ times), we can represent the formula for $\mathcal{I}^{(k)}(\boldsymbol{x}_i, y_i)$ using Bell polynomials (Ma (2013)) as follows:

$$\mathcal{I}_\theta^{(k+1)}(\boldsymbol{x}, y) = -H_\theta^{-1}\Big(\sum_{m=1}^{k} B_{m,k+1}\Big(\big\{\mathcal{I}_\theta^{(j)}(\boldsymbol{x}, y), \nabla_\theta^j \ell(y, f(\boldsymbol{x}, \theta))\big\}_{j=1}^{m}\Big) +$$

$$\sum_{m=1}^{k} B_{m,k}\Big(\big\{\mathcal{I}_\theta^{(j)}(\boldsymbol{x}, y), \nabla_\theta^j L(\mathcal{D}, \theta)\big\}_{j=1}^{m}\Big), \tag{B.14}$$

where $B_{m,k}$ is the Bell polynomial, which is defined as follows:

$$B_{m,k}(x_1, x_2, \ldots, x_{m-k+1}) = \sum \frac{m!}{j_1! j_2! \cdots j_{m-k+1}!}\left(\frac{x_1}{1!}\right)^{j_1}\left(\frac{x_2}{2!}\right)^{j_2}\cdots\left(\frac{x_{m-k+1}}{(m-k+1)!}\right)^{j_{m-k+1}}.$$

## C PROOF OF THEOREM 2

We start by proving the coverage guarantee in Section C.1 and then prove the discrimination guarantee in Section C.2.

### C.1 FREQUENTIST COVERAGE

In what follows, we analyze the coverage probability of the exact DJ confidence intervals. The proof parallels that of (Barber et al. (2019b)). Recall that the empirical $(1 - \alpha)$ quantile is defined by:

$$\widehat{Q}_{\alpha,n}(\{X_1, \dots, X_n\}) := \text{ the } \lceil (1 - \alpha)(n + 1) \rceil \text{-th smallest value of } \{X_1, \dots, X_n\}.$$

Let the test point be $(\boldsymbol{x}_{n+1}, y_{n+1})$ and suppose we have oracle access to the true test label $y_{n+1}$. For any two distinct indices $i, j \in \{1, \dots, n+1\}$ with $i \neq j$, let $\tilde{f}_{-(i,j)}$ be the model fitted on the augmented dataset $\{(\boldsymbol{x}_i, y_i)\}_{i=1}^n \cup \{(\boldsymbol{x}_{n+1}, y_{n+1})\}$, i.e., training and test data, with points $i$ and $j$ removed. Note that $\tilde{f}_{-(i,j)} = \tilde{f}_{-(j,i)}$ for any $i \neq j$, and that $\tilde{f}_{-(i,n+1)} = \widehat{f}_{-i}$ for any $i = 1, \dots, n$.

Next, we define a matrix $\boldsymbol{K} \in \mathbb{R}^{(n+1) \times (n+1)}$ with entries $\boldsymbol{K} = [K_{ij}]_{i,j}$ defined as follows:

$$K_{ij} = \begin{cases} \infty, & i = j, \\ |y_i - \tilde{f}_{-(i,j)}(\boldsymbol{x}_i)|, & i \neq j, \end{cases} \tag{C.1}$$

where the off-diagonal entries represent the residual for the $i^{th}$ point when both $i$ and $j$ are left out of model training, in addition to the difference between the model predictions when trained on the entire training dataset and when point $i$ is left out. Let the set $\mathcal{Q}$ be defined as:

$$\mathcal{Q}(\boldsymbol{K}) = \left\{ i \in \{1, \dots, n+1\} : \sum_{j=1}^{n+1} \mathbf{1}_{\{K_{ji} < K_i^\alpha\}} \geq (1 - \alpha)(n + 1) \right\}, \tag{C.2}$$

where $K_i^\alpha = \widehat{Q}_{\alpha,n}(\{K_{ij}\}_{j=1}^{n+1})$. Before proceeding, we first prove the following Lemma.

**Lemma 1.** Let $\widehat{C}_{n,\alpha}(\boldsymbol{x}_{n+1})$ be the DJ confidence interval. Then coverage condition is:

$$y_{n+1} \notin \widehat{C}_{n,\alpha}(\boldsymbol{x}_{n+1}) \Rightarrow n + 1 \in \mathcal{Q}(\boldsymbol{K}).$$

*Proof.* Suppose that $\widehat{C}_{n,\alpha}(\boldsymbol{x}_{n+1})$ does not cover the true test label $y_{n+1}$, i.e., $y_{n+1} \notin \widehat{C}_{n,\alpha}(\boldsymbol{x}_{n+1})$, then this implies that either

$$y_{n+1} > \hat{f}(\boldsymbol{x}_{n+1}) + \widehat{Q}_{\alpha,n}^+ (\{r_i\}_{i=1}^n) + \widehat{Q}_{\alpha,n}^+ (\{v_i(\boldsymbol{x}_{n+1})\}_{i=1}^n),$$

which means that $y_{n+1} > \hat{f}(\boldsymbol{x}_{n+1}) + \widehat{Q}_{\alpha,n}^+ (\{r_i\}_{i=1}^n) + v_j$ for at least $(1 - \alpha)(n + 1)$ many indices $j \in \{1, \dots, n\}$, or

$$y_{n+1} < \hat{f}(\boldsymbol{x}_{n+1}) - \widehat{Q}_{\alpha,n}^+ (\{r_i\}_{i=1}^n) - \widehat{Q}_{\alpha,n}^+ (\{v_i(\boldsymbol{x}_{n+1})\}_{i=1}^n),$$

which means that $y_{n+1} < \hat{f}(\boldsymbol{x}_{n+1}) - \widehat{Q}_{\alpha,n}^+ (\{r_i\}_{i=1}^n) - v_j$ for at least $(1 - \alpha)(n + 1)$ many indices $j \in \{1, \dots, n\}$. In either case, then, we have

$$(1 - \alpha)(n + 1) \leq \sum_{j=1}^n \mathbf{1}\left\{ y_{n+1} \notin \hat{f}(\boldsymbol{x}_{n+1}) \pm \left( \widehat{Q}_{\alpha,n}^+ (\{r_i\}_{i=1}^n) + v_j \right) \right\}$$

$$= \sum_{j=1}^n \mathbf{1}\left\{ |y_{n+1} - \hat{f}_{-j}(\boldsymbol{x}_{n+1})| > \widehat{Q}_{\alpha,n}^+ (\{r_i\}_{i=1}^n) + v_j \right\}.$$

By nothing that

$$|y_{n+1} - \hat{f}_{-j}(\boldsymbol{x}_{n+1})| = K_{n+1,j},$$

and $\widehat{Q}_{\alpha,n}^{+}\left(\{r_i\}_{i=1}^{n}\right) + v_j = K_{n+1}^{\alpha}$, we have

$$(1-\alpha)(n+1) \leq \sum_{j=1}^{n} \mathbf{1}\left\{K_{n+1,j} > K_{n+1}^{\alpha}\right\},$$

which implies that $n+1 \in \mathcal{Q}(\boldsymbol{K})$, and hence

$$\mathbb{P}\left\{y_{n+1} \notin \widehat{C}_{n,\alpha}(\boldsymbol{x}_{n+1})\right\} \leq \mathbb{P}\left\{n+1 \in \mathcal{Q}(\boldsymbol{K})\right\}.$$

which concludes the proof of the Lemma. $\qquad\square$

Given the result of Lemma 1, we use the exchangeability of the training and test points to further bound the probability $\mathbb{P}\left\{y_{n+1} \notin \widehat{C}_{n,\alpha}(\boldsymbol{x}_{n+1})\right\}$ as follows. Since the training and testing points are exchangeable, we have $\boldsymbol{K} \stackrel{d}{=} \Pi\,\boldsymbol{K}\,\Pi^{T}$ for any $(n+1)\times(n+1)$ permutation matrix $\Pi$, where $\stackrel{d}{=}$ denotes equality in distribution. Thus, we have

$$\mathbb{P}\left\{n+1 \in \mathcal{Q}(\boldsymbol{K})\right\} = \mathbb{P}\left\{j \in \mathcal{Q}(\boldsymbol{K})\right\},$$

for all $j \in \{1, \ldots, n\}$. Based on the above, we can then evaluate the coverage probability using the equivalent condition in Lemma 1 as follows:

$$\mathbb{P}\left\{n+1 \in \mathcal{Q}(\boldsymbol{K})\right\} = \frac{1}{n+1}\sum_{j=1}^{n+1}\mathbb{P}\left\{j \in \mathcal{Q}(\boldsymbol{K})\right\}$$

$$= \frac{1}{n+1}\mathbb{E}\left[\sum_{j=1}^{n+1}\mathbf{1}\left\{j \in \mathcal{Q}(\boldsymbol{K})\right\}\right]$$

$$= \frac{1}{n+1}\mathbb{E}\left[\,|\mathcal{Q}(\boldsymbol{K})|\,\right]. \tag{C.3}$$

Finally, we compute the expected size of the set $\mathcal{Q}(\boldsymbol{K})$ as follows. By combining the original coverage condition in (C.2) together with the equivalence result in Lemma 1, we have

$$(1-\alpha)(n+1) \leq \sum_{j=1}^{n+1}\mathbf{1}\left\{K_{ji} < K_i^{\alpha}\right\} \leq \sum_{j=1}^{n+1}\mathbf{1}\left\{j \notin \mathcal{Q}(\boldsymbol{K})\right\} = n+1 - |\mathcal{Q}(\boldsymbol{K})|,$$

which implies that $|\mathcal{Q}(\boldsymbol{K})| \leq \alpha(n+1)$. By combining this with (C.3), we arrive at the following inequality

$$\mathbb{P}\left\{y_{n+1} \notin \widehat{C}_{n,\alpha}(\boldsymbol{x}_{n+1})\right\} \leq \alpha.$$

The result of the Theorem follows from the fact that the approximate DJ confidence interval $\widehat{C}_{n,\alpha}^{(m)}(\boldsymbol{x}_{n+1})$ converges in probability to the exact interval as $m \to \infty$.

## C.2 DISCRIMINATION

Recall that the exact DJ interval width is given by:

$$W(\widehat{\mathcal{C}}_{\alpha,n}^{(\infty)}(\boldsymbol{x};\hat{\theta})) = 2\,\widehat{Q}_{\alpha,n}(\mathcal{R}_n) + 2\,\widehat{Q}_{\alpha,n}(\mathcal{V}_n(\boldsymbol{x})). \tag{C.4}$$

Since the term $\widehat{Q}_{\alpha,n}(\mathcal{R}_n)$ is constant for any $\boldsymbol{x}$, discrimination boils down to the following condition:

$$\mathbb{E}[\,\widehat{Q}_{\alpha,n}(\mathcal{V}_n(\boldsymbol{x}))\,] \geq \mathbb{E}[\,\widehat{Q}_{\alpha,n}(\mathcal{V}_n(\boldsymbol{x}'))\,] \Leftrightarrow \mathbb{E}[\,\ell(y, f(\boldsymbol{x};\hat{\theta}))\,] \geq \mathbb{E}[\,\ell(y', f(\boldsymbol{x}';\hat{\theta}))\,]. \tag{C.5}$$

Note that to prove the above, it suffices to prove the following:

$$\mathbb{E}[\,v_i(\boldsymbol{x})\,] \geq \mathbb{E}[\,v_i(\boldsymbol{x}')\,] \Leftrightarrow \mathbb{E}[\,\ell(y, f(\boldsymbol{x};\hat{\theta}))\,] \geq \mathbb{E}[\,\ell(y', f(\boldsymbol{x}';\hat{\theta}))\,]. \tag{C.6}$$

If the model is stable (based on the definition in (Bousquet & Elisseeff (2002))), then a classical result by (Devroye & Wagner (1979)) states that:

$$\mathbb{E}[\,|\ell(y, f(\boldsymbol{x};\hat{\theta})) - \ell_n(y, f(\boldsymbol{x};\hat{\theta}))|^2\,] \approx \mathbb{E}[\,|\ell(y, f(\boldsymbol{x};\hat{\theta})) - \ell(y, f(\boldsymbol{x};\hat{\theta}_{-i}))|^2\,] + Const., \tag{C.7}$$

as $n \to \infty$, where $\ell_n(.)$ is the empirical risk on the training sample, and the expectation above is taken over $y \mid \boldsymbol{x}$. From (C.7), we can see that an increase in the LOO risk $\ell(y, f(\boldsymbol{x};\hat{\theta}_{-i}))$ implies an increase in the empirical risk $\ell_n(y, f(\boldsymbol{x};\hat{\theta}))$, and vice versa. Thus, for any two feature points $\boldsymbol{x}$ and $\boldsymbol{x}'$, if $v(\boldsymbol{x})$ is greater than $v(\boldsymbol{x}')$, then on average, the empirical risk at $\boldsymbol{x}$ is greater than that at $\boldsymbol{x}'$.

## D    EXPERIMENTAL DETAILS

In this Section, we provide more details for the experimental setups in Section 5, in addition to further experimental results in the context of active learning and out-of-distribution detection.

### D.1    CODE SNIPPETS FOR THE DJ PROCEDURE

The DJ procedure is very simple to implement. The key component of its implementation is the computation of HOIFs. Let `model` be an instance of the `nn` module of `Pytorch`, with a set of parameters `model.parameters()` that correspond to $\hat{\theta}$ in our formulation. Below is a Python code snippet for computing the first-order influence function of `model`.

```python
def influence_function(model, training_data_indexes):
    IF   = []
    H    = hessian(model.loss, model.parameters(), create_graph=True)
    Lamb = 1e-3
    Hinv = torch.inverse(H + Lamb * torch.eye(H.shape[0]))

    for u in range(len(training_data_indexes)):
        loss = model.loss fn(model.y[training_data _indexes[u]],
        model.hidden(model.X[training_data_indexes[u], :]).float())
        grads = torch.autograd.grad(loss, model.parameters())
        IF.append(torch.mv(Hinv, stack_torch_tensors(grads)))

    return IF
```

Computing the HOIFs can be easily implemented by recursively evaluating the gradients and lower order influence function as stated in Theorem 1. Once all the ingredients of the HOIFs are evaluated, then the perturbed parameters can be evaluated as explained in the paper, and a modified LOO model is created for every sample in the training data.

### D.2    IMPLEMENTATION OF BASELINES

In what follows, we provide details for the implementation and hyper-parameter settings for all baseline methods involved in Section 5.

**Probabilistic backpropagation (PBP).** We implemented the PBP method proposed in (Hernández-Lobato & Adams (2015)) with inference via expectation propagation using the `theano` code provided by the authors in (`github.com/HIPS/Probabilistic-Backpropagation`). Training was conducted via 40 epochs.

**Monte Carlo Dropout (MCDP).** We implemented a `Pytorch` version of the MCDP method proposed in (Gal & Ghahramani (2016)). In all experiments, we tuned the dropout probability using Bayesian optimization to optimize the AUC-ROC performance on the training sample. We used 1000 samples at inference time to compute the mean and variance of the predictions. The credible intervals were constructed as the $(1 - \alpha)$ quantile function of a posterior Gaussian distribution defined by the predicted mean and variance estimated through the Monte Carlo outputs. Similar to the other baselines, we conducted training via 40 epochs for the SGD algorithm.

**Bayesian neural networks (BNN).** We implemented BNNs with inference via stochastic gradient Langevin dynamics (SGLD) (Welling & Teh (2011)). We initialized the prior weights and biases through a uniform distribution over $[-0.01, 0.01]$. We run 40 epochs of the SGLD inference procedure and collect the posterior distributions to construct the credible intervals.

**Deep ensembles (DE).** We implemented a `Pytorch` version of the DE metho (without adversarial training)d proposed in (Lakshminarayanan et al. (2017)). We used the number of ensemble members $M = 5$ as recommended in the recent study in (Ovadia et al. (2019)). Predictions of the different ensembles were averaged and the confidence interval was estimated as 1.645 multiplied by the empirical variance for a target coverage of 90%. We trained the model through 40 epochs.

**Resampling uncertainty estimation (RUE).** We implemented RUE (Schulam & Saria (2019)) in `Pytorch` using the same Hessian computation routines involved in implementing our method. We tuned the dampening factor $\lambda$ in the stabilized Hessian $H + \lambda I$ used for matrix inversion. We used 20 bootstrapped sub-samples of the training data to compute the prediction variance.

### D.3 FURTHER RESULTS ON SYNTHETIC DATA

The minimum number of HOIFs $m^*$ required to achieve the best possible AUC-ROC for the quantitative experiment ($d = 20$) in Section 5.1 is shown in Figure 5. As expected, more complex models will generally require more high order terms of the von Mises expansion in order to recover the parameters of the LOO re-trained model.

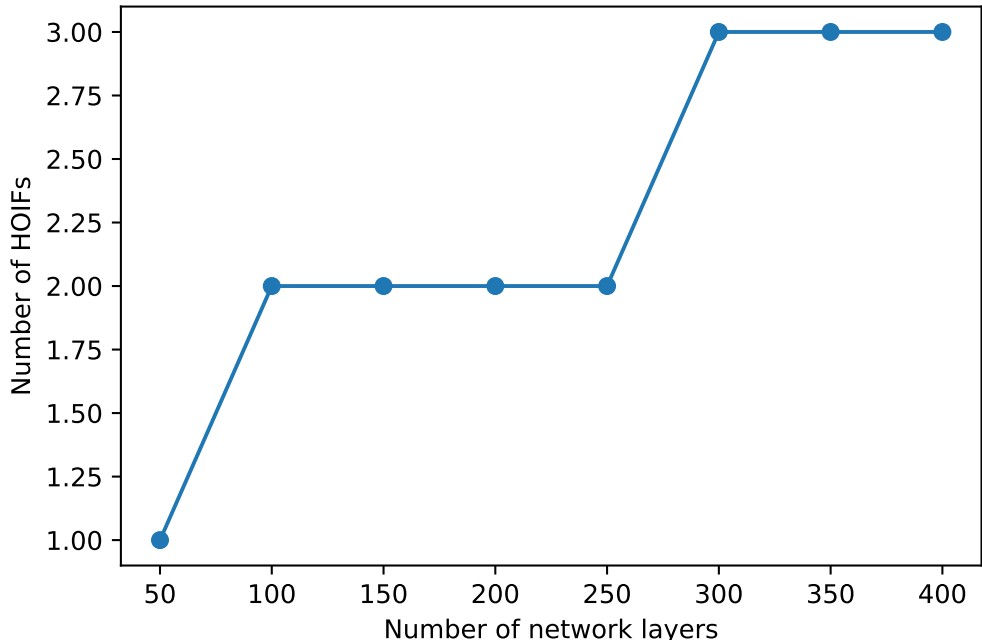

Figure 5: Impact of model complexity on number of HOIFs required for accurate recovery of LOO parameters.

### D.4 FURTHER EXPERIMENTS WITH REAL DATA

We performed another series of experiments to evaluate the accuracy of the estimates of the uncertainty estimates produced by different methods. We study two particular applications of uncertainty quantification: active learning and out-of-distribution detection (OOD).

**Active Learning.** For this setup, we replicate the active learning experiment in (Hernández-Lobato & Adams (2015)). In active learning, it is necessary to produce accurate estimates of uncertainty in order to decide which data samples we should actively collect. In this experiment, we used a neural network with a single hidden layer and 50 hidden units. We split each data set into training and test sets with 20 and 100 data instances, respectively, and pool sets with all the remaining data. After this, one data point is collected from the pool set and then moved into the training set. The process repeats until 9 of these active additions to the training set have been completed, that is, until we have performed 10 evaluations on the test set. The entire process, including the random data set splitting, is repeated 40 times. The pool data is initially lacking the target variables and these become available only once the data is moved to the training set.

We compare a baseline neural network model trained on random selections from the pool data with version of the model trained by data collected actively (PBP-A, MCDP-A, DE-A, BNN-A and DJ-A). Results are provided in Table 3. As we can see, DJ-based active collection of data is significantly better than the random baseline model on all datasets. Moreover, data actively collected using DJ-based uncertainty estimates were superior to all other methods on 6 out of the 8 datasets.

| Method | Dataset | | | | | | | |
|---|---|---|---|---|---|---|---|---|
| | Housing | Concrete | Energy | Kin8nm | Naval | Power | Wine | Yacht |
| **Baseline** | $7.0 \pm 0.7$ | $13 \pm 0.4$ | $4.5 \pm 0.2$ | $0.4 \pm 0.0$ | $0.01 \pm 0.0$ | $5.5 \pm 0.0$ | $0.95 \pm 0.1$ | $5.5 \pm 0.4$ |
| **DJ-A** | $4.8 \pm 0.3$ | $12.7 \pm 0.3$ | $3.9 \pm 0.2$ | $0.3 \pm 0.0$ | $0.01 \pm 0.0$ | $5.1 \pm 0.0$ | $0.92 \pm 0.0$ | $4.0 \pm 0.3$ |
| **PBP-A** | $5.5 \pm 0.2$ | $12 \pm 0.2$ | $4.1 \pm 0.1$ | $0.4 \pm 0.0$ | $0.02 \pm 0.0$ | $5.0 \pm 0.1$ | $0.90 \pm 0.0$ | $4.5 \pm 0.4$ |
| **MCDP-A** | $6.5 \pm 0.1$ | $12.4 \pm 0.2$ | $4.7 \pm 0.2$ | $0.5 \pm 0.0$ | $0.03 \pm 0.0$ | $5.2 \pm 0.1$ | $1.00 \pm 0.0$ | $5.1 \pm 0.4$ |
| **DE-A** | $4.2 \pm 0.2$ | $13.1 \pm 0.2$ | $4.2 \pm 0.1$ | $0.3 \pm 0.0$ | $0.05 \pm 0.0$ | $5.4 \pm 0.1$ | $0.80 \pm 0.0$ | $5.1 \pm 0.4$ |
| **BNN-A** | $7.1 \pm 0.1$ | $10 \pm 0.1$ | $4.7 \pm 0.1$ | $0.5 \pm 0.0$ | $0.02 \pm 0.0$ | $5.3 \pm 0.1$ | $1.90 \pm 0.0$ | $4.7 \pm 0.4$ |

Table 3: Average test RMSE and standard errors in active learning.

**Out-of-distribution detection.** Next, we evaluate the accuracy of the DJ uncertainty estimates in detecting whether a new test example came from the same distribution as the training data, or from another unknown distribution. This task is known as Out-of-distribution detection (OOD), and is commonly used to benchmark uncertainty estimation methods (Ovadia et al. (2019)).

In this experiment, we use data from the Infant Health and Development Program (IHDP) (Alaa & van der Schaar (2017)). The IHDP is an interventional program intended to enhance the health of premature infants. This dataset comprises observational data with treatment assignments from a real-world clinical trial, and introduced selection bias to the data artificially by removing a subset of the patients. Every sample in the dataset is of the form $(\boldsymbol{x}, y, w)$, where $w \in \{0, 1\}$ is a binary treatment assignment variable which is equal to 1 if the patient was treated, and is 0 otherwise. Since patients are not treated at random, i.e., the likelihood of a patient getting treated depends on their feature $\boldsymbol{x}$, the treated and untreated patient populations have different distributions, i.e., we have $\mathbb{P}(\boldsymbol{x} \mid w = 0) \neq \mathbb{P}(\boldsymbol{x} \mid w = 1)$. The entire dataset comprised 747 subjects, with 25 features for each subject, of which 165 were treated and the rest were untreated.

| DJ | PBP | MCDP | DE | BNN | RUE | CCP |
|---|---|---|---|---|---|---|
| $0.78 \pm 0.01$ | $0.69 \pm 0.03$ | $0.72 \pm 0.01$ | $0.75 \pm 0.02$ | $0.65 \pm 0.01$ | $0.75 \pm 0.01$ | $0.72 \pm 0.02$ |

Table 4: AUC-ROC accuracy of all baselines on the OOD task.

We conducted our experiment as follows. We train the model on 100 samples for treated patients, and leave out 65 samples for testing. In testing, we use the interval width to predict whether the test samples comes from the 65 treated patient (whose features resemble the training data), or from the untreated patients who were not involved in training. We repeated the experiment 20 times with random train/test splitting. The model we used is a feed-forward neural network with 2 hidden layers and 100 hidden units. AUC-ROC performance is reported in Table 4.

We found that DJ outperforms other methods in detecting the OOD samples, with DE and RUE being the most competitive baselines. This result is of practical significance since it implies that we can accurately determine which patients should be treated based on the recommendations of a machine learning-based decision-support system, and which patients should be examined in a case by case basis through experts. Moreover, by determining which samples in an observational dataset have the most unreliable predictions, and consequently determine what features are required in a prospective design of a clinical trial.

