# OpenReview forum: "The Discriminative Jackknife: Quantifying Uncertainty in Deep Learning via Higher-Order Influence Functions"
_ICLR.cc/2020/Conference — Reject_

### Official Review · AnonReviewer3 · 2019-10-21
**Official Blind Review #3**

**Rating:** 3

**Review:**

The authors propose using influence functions to efficiently estimate pointwise confidence intervals for regression models. Their central idea is to construct a confidence interval around a point $x$ as a function of how much the model $f$ changes around $x$ when individual training samples are left out. Their technical innovation is to combine a marginal error term that does not depend on $x$ (which ensures coverage) with a local variability error term that does depend on $x$ (to allow for greater variability in areas where the model is more uncertain and there is less data). The authors provide experimental support for the superiority of their method (in terms of coverage and discrimination) as well as theoretical support for consistency.

In my opinion, the ideas in the paper are exciting; the paper is clear and well-written; and the experimental evaluation is quite comprehensive in terms of baselines. However, I have concerns that prevent me from recommending an accept at this time:

1) Similar recursive formulations for HOIFs have appeared in the literature, so the claims in Section 3.3 should be toned down. See for example lemma 3 in Giordano, Jordan, and Broderick, 2019 (https://arxiv.org/abs/1907.12116); or for an older reference that’s a bit more specialized, see Debruyne, Hubert, and Suykens, 2008 (http://www.jmlr.org/papers/volume9/debruyne08a/debruyne08a.pdf).

2) The paper refers repeatedly to how their proposed method can be applied to deep learning models and, in particular, state-of-the-art deep learning models. In my opinion, the evidence in the paper does not support these claims. The largest experiments are run on neural networks with 100 hidden units, and it is not clear how to scale up their method to state-of-the-art models and large datasets. In particular, the Hessian-vector computations and the need to iterate through the dataset scale poorly with model and dataset size, and efficient approximations for these are non-trivial and an area of active research.

3) Related to the above point, the theorem statements apply to general differentiable loss functions. Are they true in such a general setting? For example, in Section 3.3, the assumption that the inverse Hessian needs to be positive definite is noted but this does not appear in the theorem statements. More importantly, it seems like some notion of strong convexity is required; for example, does the convergence of the von Mises series in Appendix A require that the parameters don’t change too much with $\epsilon$? This might not be true in a non-convex model. The paper also elides the fact that the global minimizer $\hat{theta}$ cannot in general be computed in non-convex models like neural networks.

The paper is otherwise compelling to me, and I believe that the above points can be remedied by being more careful and circumspect with the claims in the paper.

**Experience Assessment:**

I have published one or two papers in this area.

**Review Assessment: Checking Correctness Of Derivations And Theory:**

I assessed the sensibility of the derivations and theory.

**Review Assessment: Checking Correctness Of Experiments:**

I assessed the sensibility of the experiments.

**Review Assessment: Thoroughness In Paper Reading:**

I read the paper thoroughly.

---

> ### Author Response · Authors · 2019-11-11
> **Response to Review #3**
>
>
>
> Thank you very much for the thoughtful and very helpful comments! Much appreciated. In the submitted revised manuscript, we followed many of the suggestions put forward by the reviewer and altered the claims and discussions in the paper accordingly. We also edited the title by removing the "deep learning" key word to reinforce the fact that our focus is to develop a general conception framework for uncertainty quantification, and to acknowledge its potential computational limitations in models with many parameters.
>
> A1. Thank you for pointing out to the very recent work by Giordano et al and the classical work on model selection by Debruyne et al. We acknowledged both papers and toned down the claims in Sec 3.3 in the submitted revised manuscript. After checking both papers, we found that Giordano et al work is closer to ours, whereas Debruyne et al. has more specialized results specific to kernel-based regression models. Note though that our main goal was to develop a method for estimating point-wise confidence intervals for ML models, with influence functions being a tool used towards that end. In this sense, computation of influence functions was an intermediate step towards our main goal and not itself an end goal, hence we believe that the existence of previous attempts to compute HOIFs does not undermine our contribution.
>
> A2. In our experiments, we used the same familiar experimental settings used in previous papers on uncertainty estimation and hence the usage of networks with 100 hidden units.
>
> Scaling with respect to the number of features was evaluated in Figure 4 --- there we show that DJ both maintains the target coverage guarantee and provides better coverage with respect to other methods. With respect to scaling with data size, we believe that this should not be a bottleneck since our method needs to compute n influence functions for a dataset of size n --- for all of the n influence functions, most of the complex computations (inverse of the Hessian and loss gradients) are shared across all points, and there is not bottleneck operation that is done iteratively for each data point.  This in fact is the strength of our method; the alleviation of the complexity arising from iterative re-training in the conventional Jackknife method.
>
> Scaling with the number of parameters is definitely a crucial issue since the exact inversion of the Hessian operation is cubic in the number of parameters. However, there exists various solutions to this problem, including the stochastic approach for computing Hessian-vector products proposed by Agarwal et al. (2016) (in “Second order stochastic optimization in linear time”) and highlighted in (Koh & Liang (2017)), which can evaluate Hessian-vector products in linear time. However, since our experiments were focused on models with a relatively small number of parameters, we highlighted this issue in the revised submission, where we edited the title and other parts of the manuscript to make it clear that what we propose is a general conceptual framework for evaluating predictive uncertainty, and further investigation into its computational aspects in very “deep” models is needed.
>
> A3. Yes, our Theorem holds under the following conditions: (a) the loss function is differentiable, (b) H^-1 needs to be positive definite, and (c) the loss function must strongly convex in the local neighborhood of the local optimum \theta^hat. Assumptions (b) and (c) were stated earlier before the Theorem statement and within the proof but where missing in the main Theorem statement: we fixed this in the revised submission. We do not require though that the global optimal is achieve, our results hold for any local optimum \theta^hat.

---

### Official Review · AnonReviewer2 · 2019-10-24
**Official Blind Review #2**

**Rating:** 6

**Review:**

The authors provide an interesting study on uncertainty estimation for deep learning for regression problems. The submission is well written and well structured w.r.t. the motivation and underlying theory, which are accompanied by extensive derivations/proofs in the annexes. However, experiments are presented for small-scale tasks only.

The title of the submission addresses deep learning in general. However, the submission specifically is limited to regression, which is only mentioned scarcely throughout the paper and only as early as Sec. 2. This limitation should be mentioned in title and abstract.  Also, even though deep learning is addressed and covered in principle by the theory, the experiments do not really seem to cover deep models, being limited to two-layer networks of limited size. In Annex D.3 and specifically Fig. 5, network depth is addressed, but virtually no further details or performance measures are given for these experiments.

Many of today's deep learning tasks go beyond simple classification and handle complex tasks based on e.g. sequential or higher dimensional data strongly exploiting context, being trained on huge amounts of data, and exploiting considerable deep models with huge numbers of parameters. The experiments presented here are limited to fairly small tasks and models of limited size and especially limited depth. It would be interesting to see a discussion on how the proposed approach is expected to scale w.r.t. the size of the data set and w.r.t. the complexity of the modeling. Also, it would be interesting, to understand how prediction in context would alter the described findings.



**Experience Assessment:**

I do not know much about this area.

**Review Assessment: Checking Correctness Of Derivations And Theory:**

I assessed the sensibility of the derivations and theory.

**Review Assessment: Checking Correctness Of Experiments:**

I assessed the sensibility of the experiments.

**Review Assessment: Thoroughness In Paper Reading:**

I read the paper at least twice and used my best judgement in assessing the paper.

---

> ### Author Response · Authors · 2019-11-11
> **Response to Review #2**
>
>
>
> Thank you very much for the valuable comments.
>
> A1. We highlighted the fact that our focus is on the regression task in a more prominent place (abstract) in the revised manuscript (please refer to the submitted revision). Extension to classification is methodologically straightforward: the same approach can be exactly applied to binary or multiclass outcomes; however, we believe that the potential conflation between the notions of “uncertainty” and “calibration” in the classification setup requires a careful dedicated analysis in a separate paper. We used a two-layer network in our experiment in order to match the experiments that used the same UCI datasets in previous works on uncertainty --- the size of the network did not have much impact on the empirical results and do not affect the theoretical results. In the final manuscript, we will re-generate Table 1 for larger networks and add these results to the Appendix to highlight the fact that our results are robust for a larger number of model parameters.
>
> A2. Thank you for this suggestion. We added a discussion on how our method scales with the data size and model complexity in the submitted revision (Section 3.3). Note though that one of our key contributions is that we are able to obtain the LOO-perturbed parameters without the need for model re-training, which enables our method to scale well with the number of training examples compared to exact Jackknife methods. From Theorem 1, we can see that computing IFs requires only the loss gradients (which can be computed efficiently), and the inverse Hessian, which is computed only once for all data points, and its computational burden depends only on the number of model parameters and not the data size.
>
> Perhaps a more crucial computational bottleneck is concerned with the number of parameters. However, various methods exist for to address this issue, such as the stochastic estimation methods proposed by Agrawal et. al 2016 in "Second order stochastic optimization in linear time". We clarified this point in the revised manuscript.

---

### Official Review · AnonReviewer1 · 2019-10-28
**Official Blind Review #1**

**Rating:** 3

**Review:**

This paper studies how to construct confidence intervals for deep neural networks with guaranteed coverage. The authors propose an algorithm, “discriminative jackknife”, based on the standard jackknife confidence interval estimate which they augment by a “local uncertainty estimate” based on the variability of the n leave-one-out fitted versions of the underlying algorithm (n = # data points). The whole study is concluded with toy and real-world examples showing the proposed algorithm is competitive with existing methods while also achieving the desired coverage.

I am currently leaning towards recommending rejection of the paper. The main reasons for this are: (i) A potential failure to cite and acknowledge the prior contribution of [1] which seems to have non-trivial overlap with this paper (on arxiv since end of July which is more than 30 days before the ICLR submission deadline and thus should be treated as prior work); (ii) The claims of guaranteed frequentist coverage are not backed up as, according to thm.2, they only hold when n >> 0 and the number of influence functions used goes to infinity (ideally, the authors would provide non-asymptotic bounds as in [1], but at the very least, these limitations should have been clearly pointed out and their practical implications discussed). Finally, I would like to say that I am holding the paper to a higher standard due to its 10 page length as instructed by the guidelines.


Major comments:

- Can you please explain the relation of this work to [1]? It seems that [1] already proposes use of the higher order expansions, provides an efficient implementation based on forward mode autodiff (do you plan to release code?), and moreover provides non-asymptotic bounds which are not present in your work?!

- On a related note, Giordano et al. provide a careful analysis and discussion of the assumptions in their sect.4. Can you please clarify which of the assumptions you also make, and why you don’t need the others (if any)?

- Throughout the paper (e.g., in and around eq.1), you seem to assume that there exists a unique minimiser of the objective which generally won’t be true (especially in your application to deep neural networks). In [1], the technical assumptions ensure this is true but I don’t see how this is handled in your case?! Can you please help me understand how to interpret your work in case there are multiple (possibly local) minima, and whether this has any effect on the results in thm.2?

- I think it would be beneficial to the reader if you could please provide a discussion and/or formula for how large n has to be for Theorem 2 to apply.

- On p.8, you say “To ensure a fair comparison, the hyper-parameters of the model f(x; θ) were the same for all baselines.” I am not sure I agree this is a fair comparison. Commonly, one has the opportunity to select hyperparameters for their algorithm (e.g., using a validation set, cross-validation, etc.) so it seems it would have been fair to run each algorithm with its best hyperparameters. Can you please provide a discussion of how this would affect the reported results?


Minor comments:

- On p.3, you say “We do not pose any assumptions on how the loss function in (1) is optimized.” Do you mean to say that you do not assume anything **but** that the chosen optimiser reaches a (global?!) minimum of the loss function? It seems like you would want to exclude pathological optimisers (e.g., one that always outputs zero) but also more realistically think about the known pathologies of commonly used optimisers (see, e.g., [2]).

- Can you please clarify if and how the use of the algorithm from (Agarwal et al., 2016) for approximation of the Hessian products affects accuracy of your confidence intervals?

- In fig.3, it seems like some of the methods are not properly tuned. For example, MC-dropout should not have zero uncertainty around zero (did you by any chance set bias variance to zero?!), and BNN-SGLD does not seem to have converged (can you please provide plots providing some evidence that the MCMC sampler has mixed + information about how the hyperparameters were selected?).

- I am somewhat confused by the statement “The only hyper-parameter involved in our method is the number of HOIFs m — this was tuned by optimizing the evaluation metrics ...” on p.8. Wouldn’t thm.2 suggest that you should use as high m as possible?

- Also on p.8, can you please clarify how you selected the threshold for the evaluation of “discriminative power”? In particular, thm.2 seems to suggest that what one would desire is that the ranking based on width of predictive intervals is equivalent to the ranking based on the actual prediction error. This would suggest, for example, comparing these two rankings using Kendall’s tau coefficient (perhaps a binned modification where one would count only the number of times true error puts a point into a different bin than the width of its confidence interval). Note that I am not suggesting the above method is perfect either (it completely ignores the actual sizes of the intervals), but I’m currently having trouble interpreting the results you report, so it would be very helpful to understand how you selected this particular measure of “discriminative power” and why alternatives like the example above were discarded please.


References:

[1] Ryan Giordano, Michael I. Jordan, Tamara Broderick. A Higher-Order Swiss Army Infinitesimal Jackknife. https://arxiv.org/abs/1907.12116

[2] Ashia C. Wilson, Rebecca Roelofs, Mitchell Stern, Nathan Srebro, Benjamin Recht. The Marginal Value of Adaptive Gradient Methods in Machine Learning. https://arxiv.org/abs/1705.08292

**Experience Assessment:**

I do not know much about this area.

**Review Assessment: Checking Correctness Of Derivations And Theory:**

I did not assess the derivations or theory.

**Review Assessment: Checking Correctness Of Experiments:**

I assessed the sensibility of the experiments.

**Review Assessment: Thoroughness In Paper Reading:**

I read the paper at least twice and used my best judgement in assessing the paper.

---

> ### Author Response · Authors · 2019-11-11
> **Response to Review #1 (General comments)**
>
>
>
> Thank you very much for the valuable comments.
>
> To address concern (i), we have cited and acknowledged reference [1] in the submitted revision. Thank you for pointing us to this reference. Regarding concern (ii), we stress that the frequentist coverage is indeed guaranteed under the exact DJ procedure, and is guaranteed *for any n* under the approximate procedure as m -> infinity. In practice, m can be set to an arbitrarily high order with little computational burden, or can be set by finding the minimum m that optimizes coverage and discrimination on a validation set, hence we believe that this does not pose any serious limitation on our method.
>
> As the reviewer pointed out, non-asymptotic bounds on the accuracy of the approximate confidence intervals would be an interesting results. However, these bounds are much harder to derive compared to those in [1] because our estimand is confidence intervals on the model output and not the model parameters. Moreover, these bounds would be of little practical use since the bounds in [1] are already not tight, and --- as mentioned earlier --- m can be set to an arbitrary large order with little cost. Due to the space limitations and the already condensed content of our paper, we prefer to defer this kind of analysis to future work.

---

> > ### Author Response · Authors · 2019-11-11
> > **Response to Review #1 (Major comments)**
> >
> >
> >
> > Below are point-by-point responses to all major comments.
> >
> > *Major comments *
> >
> > A1. Yes, we will definitely release our code for computing HOIFs.
> >
> > Thank you for pointing out to the recent interesting work by Giordano et. al. We were not aware of this paper since it was only fairly recently archived. We acknowledged this paper in Section 3.3 of the submitted revision.
> >
> > Note though that the main goal of our paper was to develop a method for estimating point-wise confidence intervals for ML models, with influence functions being only a tool used towards that end. In this sense, the computation of influence function is only an intermediate step towards our main goal and not itself the end goal, hence we believe that the existence of previous attempts to compute HOIFs does not undermine our contribution.
> >
> > The non-asymptotic bounds in [1] quantify the error in the recovered parameter \theta; similar bounds on the approximation of \theta via the truncated Taylor series in (8) can be straightforwardly obtained in our framework. However, these bounds are not very useful in the problem we tackle since our goal is to obtain a confidence interval C(x; \theta) and not just estimate the LOO model parameters \hat_\theta_{-i}. Bounds on \hat_\theta_{-i} estimated via influence functions do not directly translate to bounds on the corresponding confidence interval C(x; \theta), which would depend on the specific model structure, activation functions, etc. As mentioned above, we chose to defer this analysis to future work because it would require long and heavily involved analysis with little practical significance.
> >
> > A2. All assumptions are now incorporated in the statement of Theorem 1. We will also add a separate Appendix discussing the assumptions in the final manuscript. Our analysis requires the loss function to be locally convex at the neighborhood of the local optimum --- having checked the analysis in [1], we believe that the extra assumption on uniqueness in their paper relates to the fact they study at an infinitesimal Jackknife (weighted loss function) setup and not a LOO setup as in our case.
> >
> > A3. Please note that Theorem 2 proves that the discrimination condition is satisfied asymptotically, but the **frequentist coverage is satisfied for any n**. This means that the interval width function W(C(x; \theta)) is consistent in the true uncertainty boundaries.
> >
> > Finite sample bounds on the error in discrimination function is certainly an interesting subject for analysis. Standard generalization bounds from learning theory may not apply here since we do not directly observe samples of the uncertainty intervals as in standard supervised learning.  We will add a discussion on this in a separate Appendix in the final manuscript.
> >
> > A4. A fair comparison is one that compares the quantified predictive uncertainty by different methods for the same model --- i.e., the predictions (and hence the predictive accuracy) of the underlying model must be fixed for all uncertainty quantification methods. This is why we fixed the hyper-parameters of the baseline model f(x; θ) for all baselines. Running each uncertainty quantification method with its best hyper-parameters means optimizing the predictive model itself on the accuracy of uncertainty estimates and not the accuracy of predictions --- hence the different methods would be compared in terms of the accuracy of their uncertainty estimates for different models. This is not a fair comparison since the difficulty of quantifying uncertainty will be a function of the underlying model, which in this case would be different for each baseline. For instance, imagine that the best hyper-parameters for the RUE method results in a trivial model that outputs a constant prediction of 0 for all x, and for DJ, the best hyper-parameters correspond to a model that predicts the label exactly. In this case, RUE can trivially achieve a perfect discriminative accuracy since its underlying model is uncertain about all predictions, whereas DJ will achieve an ostensibly smaller discriminative accuracy, but this is evaluated with respect to a prefect predictive model. It does not make sense in this case to say that RUE is better than DJ.
> >
> > It is also worth mentioning that we *optimized the hyper-parameters of each uncertainty method* given the fixed baseline model hyperparameters. This in our view is a fair approach because it enables each method to perform as good as possible while ensuring that all methods quantify the uncertainty of the same predictive model. We will clarify this in the final version of the manuscript.

---

> > > ### Author Response · Authors · 2019-11-11
> > > **Response to Review #1 (Minor comments)**
> > >
> > > Below are point-by-point responses to all minor comments.
> > >
> > > *Minor comments *
> > >
> > > A5. We meant by the statement on page 3 that our analysis is oblivious to the training method used as long as it retrieves a local minimum. In the final manuscript, we will be careful to make a clear exclusion of the pathological optimizer in [2]. Thank you for this remark.
> > >
> > > A6. In our experiments, approximate Hessian-vector products were accurate enough that the discriminative performance was comparable to exact re-training (and consequently exact Hessians). Whether or not this would hold in any other experimental setup is something that one needs to be cautious about --- we will add a remark on this in the final manuscript.
> > >
> > > A7. For all methods, hyper-parameters were tuned to optimize discriminative accuracy of the confidence intervals. Calibrating the MC-dropout can be cumbersome in many problems since the appropriate values of the precision parameters can be very small. We sweeped a wide range of hyperparameters to find the best tuning for the MCDP benchmark in Fig. 3. We will also provide the MCMC trace for BNN-SGLD in the Appendix.
> > >
> > > A8. We agree that calling m a "hyper-parameter" may be confusing --- m should be set as the highest value that gives the best validation performance. We will fix that in the final manuscript.
> > >
> > > A9. Thank you for this suggestion. Our experiments originally looked at all thresholds (error percentiles 1 to 100) and we compared all methods by stacking their AUC-ROC at all thresholds. Since it was impossible to plot all these curves in one page, and since the curves of our method were not crossing with other baselines at any threshold, we thought it is more convenient to look at one threshold and summarize the AUC-ROC values in a Table.
> > >
> > > We have already tried to use a C-index evaluation instead of the AUC-ROC since C-index would not rely on any threshold for binarization of the predicted uncertainties. (C-index is a concordance measure tightly related to the Kendall coefficient suggested by the reviewer.) We preferred to present AUC values since we thought our readership is more familiar with it, whereas C-index is only common in survival analysis. There were no differences in the performance trends based on AUC and C-index --- we can switch our results back to the C-index if the reviewer believes this is more appropriate.

---

### Official Review · AnonReviewer4 · 2019-11-03
**Official Blind Review #4**

**Rating:** 6

**Review:**

In this work, the authors develop the discriminative jackknife (DJ), which is a novel way to compute estimates of predictive uncertainty. This is an important open question in machine learning and the authors have made a substantial contribution towards answering the question of "can you trust a model?" DJ constructs frequentist confidence intervals via a posthoc procedure. Throughout, the authors provide excellent background and exposition. They develop an exact construction of the DJ confidence intervals in Section 3.1. This is an intuitive approach that the authors explain well. Next, they explain and then develop the concept of higher order influence functions. They do a great job of communicating this concept. Section 3.4 provides the theoretical guarantees for DJ. The related work section is extensive and thorough. The authors have thoughtful experiments that demonstrate positive attributes of DJ.

I suggest that this paper is weak accepted. On synthetic and real data, the DJ empirically works well, based on Figure 4 and Table 2. In addition, the theoretical exposition is very clear and compelling. The intuition provided in Sections 3.1 and 3.2 helps readers really understand what's going on, then 3.3 and 3.4 give theoretical justifications of the utility of DJ.

However, I have a few suggestions for improvement that lead to the "weak" acceptance. First, I'll cover minor quibbles, then more major points.

In Figure 1: At first, the blue dots and blue shading were not clear to me. In the legend, maybe explain that the blue shading indicates coverage and the blue dots indicate regions of higher/lower discrimination.

In Equation 6: Looks like there is a misplaced parentheses. I think the last two terms of the equations should be Q(Vn-)+Q(Vn+) not Q(Vn-+Q(Vn+))

In Equation 7, and throughout: I think a better notation for the function would be \mathcal{I}^{(1)}_{\hat{\theta}} rather than \mathcal{I}^{(1)}_{\theta} since the influence function is a derivative with respect to the optimal parameters.

Below equation 8: you should probably have an additional k exponent in the numerator of the kth order influence term, i.e. = \frac{\del^k \hat{\theta}_{i, \epsilon}}{\del \epsilon^k}

In Theorem 1: could you calculate this without \grad L(D, \theta), since L a function of \ell ? Maybe mention this in the appendix

It could be nice for an appendix study on how approximating the Hessian impacts performance for cases when we can compute the Hessian exactly.

Table 1 could be expanded to include comparisons on computational bottlenecks and if there's retraining in these other methods.

In Figure 4, please indicate the order of the IF used in the DJ procedure. DJ(m=?)?

Major issues:

Why weren't the other jackknife procedures used as baselines as well? I realize DJ has advantages compared to them, but an apples to apples comparison would be useful. For some researchers, LOO CV might not be prohibitive. This could be a chance to really sell your method: if it does well enough compared to more expensive LOO jackknife procedures, that would be a compelling reason to choose DJ.

Could you please check this reference and let us know if it substantial is different from Influence functions that you develop? "Higher order influence functions and minimax estimation of nonlinear functionals" Robins 2008 DOI: 10.1214/193940307000000527 Robins et al. develop a way to compute higher order influence functions, which you do claim you're the first to do "to the best of your knowledge."


**Experience Assessment:**

I have read many papers in this area.

**Review Assessment: Checking Correctness Of Derivations And Theory:**

I carefully checked the derivations and theory.

**Review Assessment: Checking Correctness Of Experiments:**

I assessed the sensibility of the experiments.

**Review Assessment: Thoroughness In Paper Reading:**

I read the paper thoroughly.

---

> ### Author Response · Authors · 2019-11-11
> **Response to Review #4 (Major issues)**
>
>
>
> Thank you for the valuable comments and suggestions.
>
> *Major issues*
>
> A1. Thank you for suggesting the inclusion of Jackknife procedures as baselines. Note that in the original manuscript, we have already compared the interval-width of DJ to that of the Jackknife-minimax procedure (Figure 4 – rightmost panel). Since Jackknife-minimax is the only other Jackknife procedure that still guarantees (1-\alpha) coverage, we thought that this makes it the only sensible baseline for an apple-to-apple comparison with our method, i.e. comparing the interval width of both methods to examine how wide the confidence intervals need to be in both methods to achieve the coverage guarantee.
>
> Because Jackknife-minimax is an overly conservative procedure, it falls short when it comes to discriminative accuracy. When applied to the experiment in Table 1, it underperformed compared to DJ (AUC-ROC: 0.85, 0.5, 0.87, 0.67, 0.85, 0.55, 0.71, 0.88 for the 8 datasets in Table 1 following the same order from left to right). Thus, the only Jackknife procedure with coverage guarantee (Jackknife-minimax) is computationally prohibitive (since it demands re-training), is overly conservative (wide confidence intervals) and is less accurate in the discrimination task. We will add these results in Table 1 in the revised manuscript.
>
> We also note that we have already included the RUE and CCP baselines in our performance comparisons. The RUE baseline is simply a bootstrapping procedure that estimates the perturbed model parameters using the inverse Hessian of its loss, and then estimates the confidence intervals as the variance of the predictions made by perturbing the weights of individual data points in the loss function. This resembles a (weighted) Jackknife procedure with first-order IF and with only the local variability term in (5) included in the confidence interval construction. (CCP follows a similar procedure but with exact re-training and a randomized construction of the confidence intervals.) Both methods under-perform in terms of coverage and discrimination as we can see in Table 1.
>
> Whenever LOO CV is not prohibitive, then an exact construction of the DJ method can be used. In this case, an apple-to-apple comparison would be an exactly computed DJ with an exactly computed RUE, and Jackknife-Minimax. We already know that when the LOO parameters are not exact, DJ outperforms RUE --- this would likely continue to be the case if parameters are exact since the reason DJ outperforms RUE is related to the way confidence intervals are constructed, and not the way LOO parameters are approximated. Also, even when LOO parameters are not exact, DJ still outperforms the exact Jackknife-minimax as mentioned earlier.
>
> In the final manuscript, we will add the results of the Jackknife-minimax to Table 1, and highlight that RUE is a surrogate for a standard Jackknife procedure that estimates confidence intervals based on prediction variance (local variability) only. In the Appendix, we will add extra results with exact LOO procedures for DJ, Jackknife-minimax, Jackknife+ and RUE to further highlight the strength of our method.
>
> A2. Thank you for pointing out to the seminal work by Robins et al. We are familiar with this great work. In this work, however, the focus was slightly different: the goal was to evaluate the estimating influence functions of various functionals common in epidemiology (average treatment effects, regression functions with missing predictors, etc) in order to derive minimax optimal estimates of these functionals. This is achieved by first obtaining a plug-in estimate of the target functional, and then correcting for the resulting plug-in bias which can be estimated using the influence functions.
>
> While Robins et al. lay out a general theory for evaluating HOIFs, our results cannot be subsumed under any of the results therein. The results in Section 3 in this paper focus on certain classes of doubly robust functionals, and obtains closed-form expressions for only the first-order IF of the expected conditional co-variance of a regression function, the variance-weighted average treatment effect, other functional in the missing data setting, etc. A kernel-based method is proposed in Section 3.2 to evaluate HOIFs, but it is motivated by the class of functionals and the problems under study in this paper, and is not directly applicable to our setup (Theorem 1 in our paper).
>
> To sum up, as far as we are concerned, the results in Robins et al are focused on deriving the efficient IFs for the sake of minimax optimal estimation of a specific class of functionals, and no closed-form expression presented therein directly applies to our setup. However, we agree that understanding the connections between this paper and ours and adding a discussion in the revised manuscript would definitely be useful. We cited this paper in the submitted revision and will add a discussion in the final manuscript.

---

> > ### Author Response · Authors · 2019-11-11
> > **Response to Review #4 (Minor issues)**
> >
> >
> >
> > *Minor issues*
> >
> > A3. In the submitted revised manuscript, we added a description in the caption of Figure 1 to clarify the significance of the blue dots and the blue shaded region.
> >
> > A4. Thank you for pointing out to the missing parentheses in Eq. 6 and the additional exponent below Eq. 8. We fixed these issues in the submitted revision. Regarding the suggested notational change in Eq. 7 (and throughout), we will incorporate this in the final manuscript.
> >
> > A5. Yes, it suffices to evaluate \grad \ell for Theorem 1 for evaluating the HOIFs. As suggested by the reviewer, we will mention this in the Appendix.
> >
> > A6. Exact Hessian matrix computation is possible for small neural networks (i.e., networks with a relatively small number of parameters). We will compare the confidence intervals obtained by the exact and approximate Hessian computation for the synthetic experiments in Section 5.1 and add these results and the relevant discussion to the Appendix.
> >
> > A7. DJ and MCDP are the methods with the fastest runtime --- we will add a column to Table 1 with the average runtime of every method to highlight this.
> >
> > A8. In Figure 4, the order of IFs is m=2. We will clarify this in the discussion of the results.

---

### Decision · Program_Chairs · 2019-12-19

**Decision:**

Reject

**Comment:**

In this work, the authors develop a method for providing frequentist confidence intervals for a range of deep learning models with coverage guarantees.  While deep learning models are being used pervasively, providing reasonable uncertainty estimates from these models remains challenging and an important open problem.  Here, the authors argue that frequentist statistics can provide confidence intervals along with rigorous guarantees on their quality.  They develop a jack-knife based procedure for deep learning.  The reviews for this paper were all borderline, with two weak accepts and two weak rejects (one reviewer was added to provide an additional viewpoint).  The reviewers all thought that the proposed methodology seemed sensible and well motivated.  Among the cited issues, major topics of discussion were the close relation to related work (some of which is very recent, Giordano et al.) and that the reviewers felt the baselines were too weak (or weakly tuned).  The reviewers ultimately did not seem convinced enough by the author rebuttal to raise their scores during discussion and there was no reviewer really willing to champion the paper for acceptance.  Unfortunately, this paper falls below the bar for acceptance.  It seems clear that there is compelling work here and addressing the reviewer comments (relation to related work, i.e. Robbins, Giordano and stronger baselines) would make the paper much stronger for a future submission.